# Gli3 utilizes Hand2 to synergistically regulate tissue-specific transcriptional networks

Kelsey H Elliott[1,2,3], Xiaoting Chen[4], Joseph Salomone[1,3,5], Praneet Chaturvedi[1], Preston A Schultz[1,2], Sai K Balchand[1,2], Jeffrey D Servetas[6], Aimée Zuniga[7], Rolf Zeller[7], Brian Gebelein[1], Matthew T Weirauch[1,4], Kevin A Peterson[6]*, Samantha A Brugmann[1,2,8]*

[1]Division of Developmental Biology, Cincinnati Children's Hospital Medical Center, Cincinnati, United States; [2]Division of Plastic Surgery, Department of Surgery, Cincinnati Children's Hospital Medical Center, Cincinnati, United States; [3]Graduate Program in Molecular and Developmental Biology, Cincinnati Children's Hospital Research Foundation, Cincinnati, United States; [4]Center for Autoimmune Genomics and Etiology, Department of Pediatrics, Cincinnati Children's Hospital Medical Center, Cincinnati, United States; [5]Medical-Scientist Training Program, University of Cincinnati College of Medicine, Cincinnati, United States; [6]Jackson Laboratory, Bar Harbor, United States; [7]Developmental Genetics, Department of Biomedicine, University of Basel, Basel, Switzerland; [8]Shriners Children's Hospital, Cincinnati, United States

*For correspondence:
kevin.peterson@jax.org (KAP);
samantha.brugmann@cchmc.org
(SAB)

**Competing interests:** The authors declare that no competing interests exist.

**Abstract** Despite a common understanding that Gli TFs are utilized to convey a Hh morphogen gradient, genetic analyses suggest craniofacial development does not completely fit this paradigm. Using the mouse model (*Mus musculus*), we demonstrated that rather than being driven by a Hh threshold, robust Gli3 transcriptional activity during skeletal and glossal development required interaction with the basic helix-loop-helix TF Hand2. Not only did genetic and expression data support a co-factorial relationship, but genomic analysis revealed that Gli3 and Hand2 were enriched at regulatory elements for genes essential for mandibular patterning and development. Interestingly, motif analysis at sites co-occupied by Gli3 and Hand2 uncovered mandibular-specific, low-affinity, 'divergent' Gli-binding motifs (**d**GBMs). Functional validation revealed these **d**GBMs conveyed synergistic activation of Gli targets essential for mandibular patterning and development. In summary, this work elucidates a novel, sequence-dependent mechanism for Gli transcriptional activity within the craniofacial complex that is independent of a graded Hh signal.

## Introduction

The Hedgehog (Hh) signaling pathway has been studied for decades in contexts ranging from organogenesis to disease (*Nüsslein-Volhard and Wieschaus, 1980*; *Chang et al., 1994*; *Chiang et al., 1996*; *St-Jacques et al., 1999*; *Hebrok et al., 2000*; *Yao et al., 2002*; *Zhang et al., 2006*). Transduction of the pathway in mammals relies on the activity of three glioma-associated oncogene (Gli) family members Gli1, 2, and 3, thought to be derived from duplications of a single ancestral gene similar to those found in lower chordates (*Shin et al., 1999*; *Shimeld et al., 2007*). While Gli2 and Gli3 transcription factors (TFs) function as both activators and repressors of Hh target genes (*Dai et al., 1999*; *Sasaki et al., 1999*; *Bai et al., 2004*; *McDermott et al., 2005*), genetic experiments have determined that Gli2 functions as the predominant activator of the pathway (*Ding et al.,*

*1998*; *Matise and Joyner, 1999*; *Park et al., 2000*), whereas Gli3 functions as the predominant repressor (*Persson et al., 2002*). All Gli family members contain five zinc-finger domains and numerous approaches (ChIP, SELEX and Protein- Binding Microarray) have confirmed each recognizes a common consensus sequence, GACCACCC as the highest affinity site (*Kinzler and Vogelstein, 1990*; *Hallikas et al., 2006*; *Vokes et al., 2007*; *Vokes et al., 2008*; *Peterson et al., 2012*). This shared consensus sequence suggests other factors and variables contribute to shaping tissue-specific and graded Gli-dependent transcriptional responses.

The fundamental and prevailing hypothesis explaining graded Hh signal transduction is the morphogen gradient (*Wolpert, 1969*). In this model, the secreted morphogen (Sonic Hedgehog; Shh) emanates from a localized source and diffuses through a tissue to establish a gradient of activity. Responding cells are hypothesized to activate differential gene expression in a concentration dependent manner, which subsequently subdivides the tissue into different cell types. Over the years, there have been edits to the original morphogen gradient hypothesis including superimposition of a temporal variable (*Dessaud et al., 2007*; *Dessaud et al., 2010*; *Balaskas et al., 2012*) and understanding how the heterogeneity in receiving cells can lead to diverse responses to the morphogen (*Jaeger et al., 2004*; *Dessaud et al., 2008*; *Balaskas et al., 2012*). However, two highly studied tissues, the developing neural tube (NT) and limb, have provided the best support and understanding for the morphogen gradient as the primary mechanism used by the Hh pathway to pattern tissues.

In the NT, Shh is produced from the ventral floor plate and forms a concentration gradient along the dorsal-ventral (DV) axis that is subsequently translated into a Gli activity gradient with Gli activator (GliA) levels higher ventrally and Gli repressor (GliR) levels higher dorsally (*Echelard et al., 1993*; *Roelink et al., 1994*; *Briscoe and Ericson, 2001*; *Wijgerde et al., 2002*). These opposing GliA and GliR gradients correlate with *Gli2* and *Gli3* expression patterns, respectively, and are required for patterning motor neurons and interneurons along the DV axis of the NT (*Lei et al., 2004*). While the most ventral cell types are lost in Gli2 mutants, Gli3 mutants have only a moderate phenotype (*Ding et al., 1998*; *Persson et al., 2002*). These observations suggest that cell identity within the ventral NT is more sensitive to levels of GliA than GliR.

In contrast, the developing limb utilizes Gli3R to perform the major patterning role, with Gli2 playing only a minor role (*Hui and Joyner, 1993*; *Mo et al., 1997*; *Bowers et al., 2012*). Shh and Gli3R form opposing gradients across the anterior-posterior (AP) axis of the limb bud. Loss of *Gli3* results in polydactyly and a partial loss of AP patterning, suggesting that a Gli3R gradient is necessary to determine digit number and polarity (*Wang et al., 2000*; *Litingtung et al., 2002*; *te Welscher et al., 2002*). Furthermore, Gli3 is epistatic to Shh: the $Shh^{-/-};Gli3^{-/-}$ compound knockout has a polydactylous limb phenotype identical to the *Gli3* mutant alone, indicating that the major role of Shh in the autopod is to modulate Gli3R formation (*Litingtung et al., 2002*; *te Welscher et al., 2002*). Thus, these classic genetic studies established the understanding that the formation of distinct Gli2 (activator) and Gli3 (repressor) gradients are necessary for converting the Hh signal transduction cascade into downstream gene expression responses within the vertebrate NT and limb.

The developing craniofacial complex represents another organ system heavily reliant upon Shh signal transduction for proper development and patterning (*Helms et al., 1997*; *Marcucio et al., 2001*; *Hu, 2003*; *Cordero et al., 2004*; *Lan and Jiang, 2009*; *Young et al., 2010*; *Xu et al., 2019*); however, the mechanisms by which the craniofacial complex translates a Shh signal remain much more nebulous than those in the NT or limb. Several issues contribute to the lack of clarity in the developing face. First, rather than the simple morphology of a tube or a paddle, the facial prominences have complex morphologies that rapidly and significantly change throughout development. Second, unlike the NT and limb, patterns of *Gli2* and *Gli3* expression are not spatially distinct within the facial prominences (*Hui et al., 1994*). For example, despite an epithelial source of Shh on the oral axis in the developing mandibular prominence (MNP), opposing gradients of *Gli2* and *Gli3* have not been reported. Finally, conditional loss of either *Gli2* or *Gli3* alone in the neural crest cell (NCC)-derived facial mesenchyme does not result in significant patterning defects indicative of a gain- or loss-of-Hedgehog function (*Chang et al., 2016*). Together, these data suggest that additional mechanisms of Gli-mediated Hh signal transduction are utilized during facial development to initiate proper patterning and growth.

In this study we combined expression, genetic, genomic and bioinformatic studies to identify a novel, Gli-driven mechanism of activating tissue-specific transcriptional networks to confer Hh-

induced positional information independent of a morphogen gradient. Specifically, Gli3 and Hand2 utilize low-affinity, divergent GBM (**d**GBM) and E-boxes to promote synergistic activation of MNP targets, outside the highest threshold of Hh signaling. We uncovered novel genetic and physical interactions between Gli3 and the bHLH TF Hand2 within the developing MNP. Genomic binding analyses highlighted enrichment of both factors at the same CRMs and revealed a surprising, motif-dependent synergism distinct to Gli3 and Hand2. Importantly, this synergism is required for robust activation of Gli targets important for mandibular patterning, glossal development and skeletogenesis. Our findings suggest that context-dependent optimization of Gli- binding site occupancy in the presence of Hand2 is critical for modulating tissue-specific transcriptional output within a tissue that lacks an obvious Shh morphogen gradient. Hence, these findings define how craniofacial prominences can serve as distinct developmental fields that interpret Hh signals in a manner unique to other organ systems.

## Results

### Loss of Gli TFs and Hand2 generates micrognathia and aglossia

To attain a comprehensive understanding of Gli TF function during craniofacial development, we generated conditional mutant mice lacking *Gli2* and *Gli3* in the NCC-derived mesenchyme (*Gli2^{f/f}; Gli3^{f/f};Wnt1-Cre,* herein referred to as *Gli2/3* cKO). While these mutants present with a variety of cranial defects including mid-facial widening, cleft lip/palate (*Chang et al., 2016*) and a domed cranial vault; we also observed a severe micrognathic phenotype in *Gli2/3* cKO embryos. Relative to wild-type embryos, *Gli2/3* cKO mutants presented with low-set pinnae, aglossia and micrognathia (*Figure 1A–C', I*). While the distal mandible was hypoplastic and certain distal structures such as the incisors were absent, the proximal mandible was more severely affected. Proximal mandibular structures such as the coronoid, condylar, and angular processes, were almost completely lost (*Figure 1D,I*, *Figure 1—figure supplement 1A–B*) and posterior cranial skeletal structures including the tympanic ring were hypoplastic. Interestingly, conditional loss of either *Gli2* or *Gli3* alone (*Gli2^{f/f}; Wnt1-Cre* or *Gli3^{f/f};Wnt1-Cre*) did not replicate the mandibular phenotype observed in double mutants (*Chang et al., 2016*).

While *Hand2^{f/f};Wnt1-Cre* mutants (herein referred to as *Hand2* cKO) did not present with mid-facial, clefting or calvarial phenotypes, they did present with low-set pinnae, aglossia and micrognathia, similar to *Gli2/3* cKO embryos (*Figure 1E–E', I*; *Morikawa et al., 2007*; *Barron et al., 2011*). Skeletal analysis of *Hand2* cKO mutants confirmed a dysmorphic and hypoplastic mandible and loss of Meckel's cartilage (*Figure 1F,I*). Compared to the *Gli2/3* cKO embryos, *Hand2* cKO embryos exhibited a less severe proximal mandibular phenotype. While the tympanic ring and angular processes were absent, the coronoid and condylar processes were not severely hypoplastic (*Figure 1—figure supplement 1C*). Although the hyoid bone was present, it was abnormally fused to middle ear cartilage and underwent excessive/ectopic ossification (*Barron et al., 2011*). Most strikingly; however, the *Hand2* cKO mutants exhibited extreme distal jaw hypoplasia. Together, these phenotypic analyses suggested that while Gli2 and Gli3 were predominantly required for proximal jaw development and Hand2 was predominantly required for distal jaw development, both Gli2/3 and Hand2 were necessary for proper mandibular development.

To determine if and how Gli2/3 and Hand2 function together during mandibular development, we first tested if there was an epistatic relationship between Gli TFs and Hand2. We analyzed gene expression in both conditional KO mutant embryos by RNA-seq and did not detect significant changes in expression of *Hand2* in *Gli2/3* cKO MNPs, or significant changes in the expression of Shh pathway components in *Hand2* cKO MNPs (*Supplementary file 1*). Thus, in contrast to the limb (*Vokes et al., 2008*), these data suggest that rather than functioning up or downstream of one another, these TFs may work in parallel to promote MNP patterning and development.

To test the hypothesis that Gli TFs and Hand2 regulate a common transcriptional network within NCCs of the MNP, we performed combinatorial genetic and biochemical experiments. First, while heterozygous *Gli2/3* or *Hand2* conditional mutants (*Gli2^{f/+};Gli3^{f/+};Wnt1-Cre* or *Hand2^{f/+};Wnt1-Cre*, respectively) did not produce significant MNP phenotypes (*Figure 1—figure supplement 1E–F'*), triple heterozygotes (*Hand2^{f/+};Gli2^{f/+};Gli3^{f/+};Wnt1-Cre*) resulted in subtle yet significant MNP phenotypes, including low-set pinnae, micrognathia, smaller incisors, and hypoglossia (*Figure 1—figure*

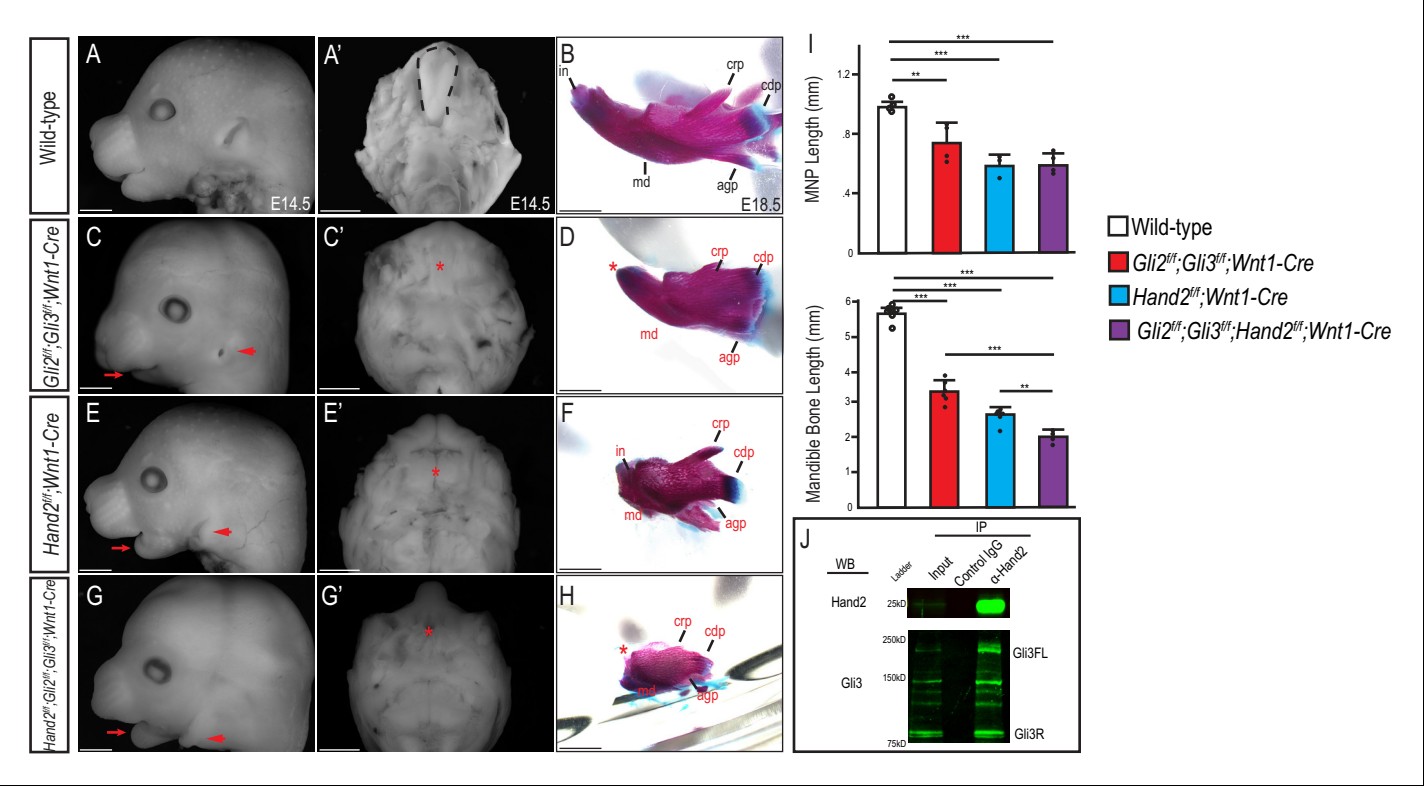

**Figure 1.** Gli and Hand2 are required for mandibular development *in vivo*. (A,C,E,G) Lateral cranial view or (A',C',E',G') dorsal mandibular view of wild-type, *Gli2^{f/f};Gli3^{f/f};Wnt1-Cre, Hand2^{f/f};Wnt1-Cre*, and *Gli2^{f/f};Gli3^{f/f};Hand2^{f/f};Wnt1-Cre* embryos at E14.5. Red arrow indicates micrognathia. Red arrowhead indicates low-set pinnae. Dotted black line denotes tongue and red asterisk highlights observed aglossia. (B,D,F,H) Lateral view of Alizarin Red and Alcian Blue staining to mark bone and cartilage respectively in wild-type, *Gli2^{f/f};Gli3^{f/f};Wnt1-Cre, Hand2^{f/f};Wnt1-Cre*, and *Gli2^{f/f};Gli3^{f/f};Hand2^{f/f};Wnt1-Cre* mandibles at E18.5. Abbreviations: md, mandible; in, incisor; crp, coronoid process; cdp, condylar process; (I) Measurements of MNP and mandibular bone. Data are expressed as mean + SD with individual data points. *p<0.05, **p<0.01, ***p<0.001. (J) Co-immunoprecipitation showing interaction between Gli3 and Hand2 within E10.5 MNPs. Scale bar: 1 mm. See also *Figure 1—figure supplement 1*.

The online version of this article includes the following source data and figure supplement(s) for figure 1:

**Source data 1.** Differences in gene expression levels from conditional KO bulk RNA-seq.

**Figure supplement 1.** Variations in Gli and Hand2 affect craniofacial development.

supplement 1G–H', I). Further, triple homozygous mutants (*Gli2^{f/f};Gli3^{f/f};Hand2^{f/f};Wnt1-Cre*) presented with the most severe and significant lower jaw phenotypes when compared to all other combinatorial mutants, including low-set pinnae, aglossia and a complete loss of both proximal and distal MNP structures (*Figure 1G–I, Figure 1—figure supplement 1D*). Thus, these genetic experiments supported the possibility that Gli TFs and Hand2 function together for proper MNP development across the full proximal-distal axis.

Finally, to determine if Hand2 and Gli TFs physically interact *in vivo*, we performed co-immunoprecipitation assays using embryonic day (E) E10.5 wild-type MNPs. Hand2 physically interacted with both full-length and truncated isoforms of Gli3, but only the truncated isoform of Gli2 (*Figure 1J, Figure 1—figure supplement 1J*). Taken together, these data provided genetic, molecular and biochemical evidence suggesting that Gli and Hand2 TFs participate within a common transcriptional network important for mandibular development, and further suggested that there may be a unique role for Gli/Hand2 cooperation.

## Gli2, Gli3, and Hand2 are co-expressed in NCC-derived populations associated with skeletal and glossal progenitors

To explore the molecular basis for Gli-mediated micrognathia and investigate the hypothesis that Gli TFs and Hand2 cooperate to initiate MNP patterning and development, we examined the

endogenous expression of these TFs during early MNP development using single molecule fluorescent *in situ* hybridization (RNAscope). Contrary to the distinct and opposing *Gli2* and *Gli3* expression domains observed in other developing organ systems (*Lee et al., 1997*; *Sasaki et al., 1997*; *Büscher and Rüther, 1998*; *Lei et al., 2004*), no spatial distinction or opposing expression gradients were observed between *Gli2* and *Gli3* in the developing MNP (*Figure 2A–C'*). Furthermore, *Gli2* and *Gli3* were co-expressed within many cells of the developing MNP (*Figure 2C–C'*), supporting the hypothesis that the developing MNP uses unique mechanisms to integrate spatiotemporal information.

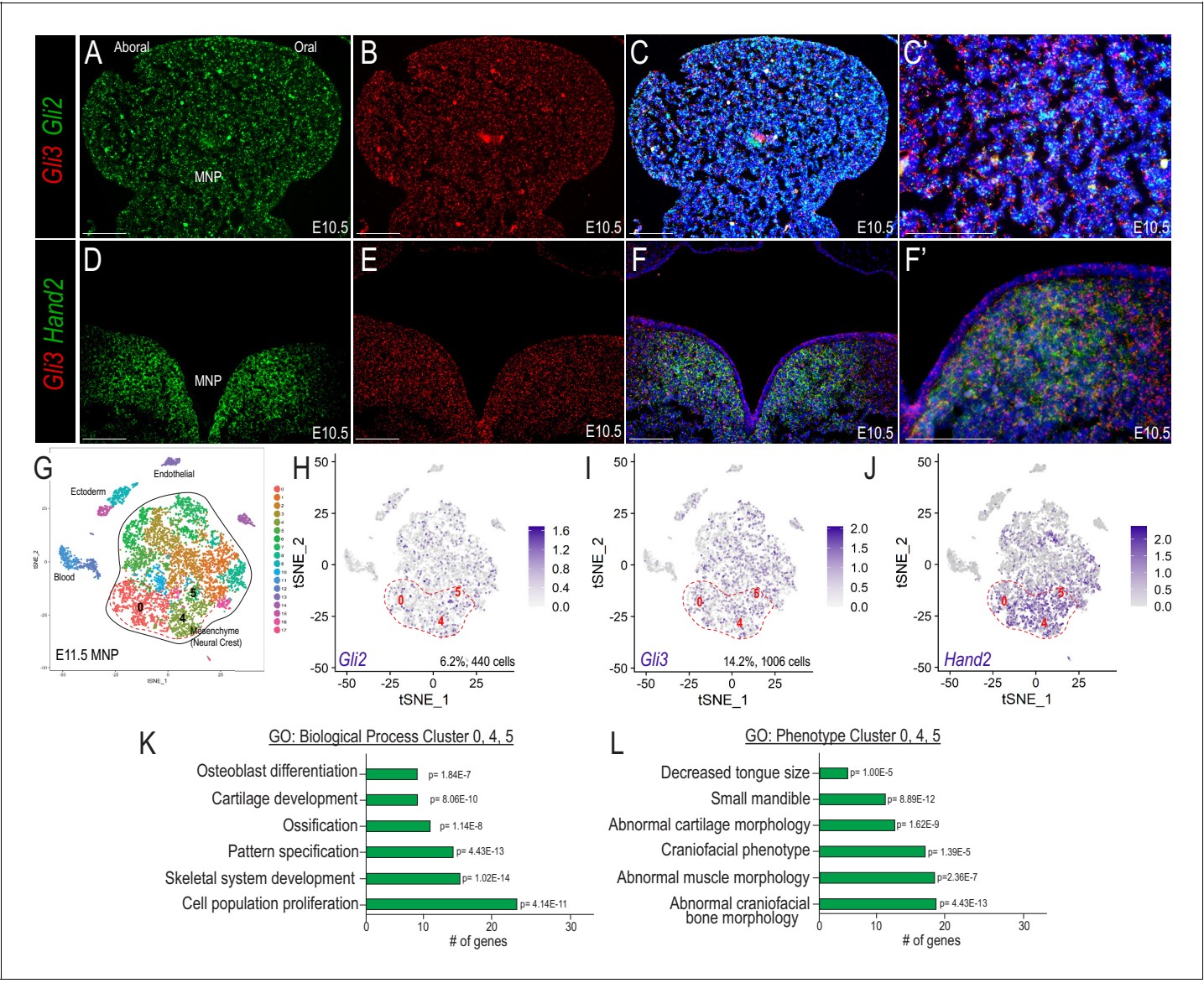

**Figure 2.** Co-expression of Gli and Hand2 in a subset of skeletal and muscle-promoting NCCs. (A–C) Expression of *Gli2* and *Gli3* within the developing MNP as revealed by smFISH on sagittal sections of E10.5 embryos. (C') Higher magnification of C. (D–F) Expression of *Gli3* and *Hand2* within the developing MNP as revealed by smFISH on frontal sections of E10.5 embryos. (F') Higher magnification of F. (G) tSNE plot of single-cell RNA-sequencing of the E11.5 MNP. (H–J) Single-cell expression of *Gli2, Gli3,* and *Hand2* in the E11.5 MNP. Dotted red line indicates *Gli+/Hand2+* NCC clusters (0, 4, 5). (K–L) GO-terms associated with marker genes for clusters 0, 4, 5 indicate *Gli+/Hand2+* cells may contribute to skeletogenesis and glossal development. Scale bar: 100 µm. See also *Figure 2—figure supplement 1*.

The online version of this article includes the following figure supplement(s) for figure 2:

**Figure supplement 1.** cNCC derivates in the early MNP.

As opposed to the widespread MNP expression of *Gli3* and *Gli2, Hand2* expression was confined to the medial aspect of the MNP (*Figure 2D–E*; *Srivastava et al., 1997*; *Thomas et al., 1998*; *Barron et al., 2011*; *Funato et al., 2016*). Interestingly, while many *Gli3+* cells did not express *Hand2*, most or all *Hand2+* cells co-express *Gli3* (*Figure 2F,F'*). To confirm co-expression and further determine the identity of cells co-expressing *Gli2/3* and *Hand2*, we performed single-cell RNA-sequencing (scRNA-seq) in the developing MNP. At E11.5, unsupervised clustering identified 17 distinct clusters in the MNP, including a central grouping of mesenchymal clusters derived from NCCs (*Figure 2G*; *Figure 2—figure supplement 1A–C*). Coincident with RNAscope, *Gli2* and *Gli3* expression were not restricted to, nor enriched in any particular cell cluster. While we failed to observe a gradient or polarized expression of Gli TFs throughout the MNP, there were over 2-fold more cells expressing *Gli3* compared to *Gli2* (*Figures 2H–I*, 1006 cells, 14.2% vs., 440 cells, 6.2%). In contrast to *Gli3* expression, *Hand2* expression was not uniformly expressed, with 43% of cells expressing *Hand2* occupying clusters 0,4 and 5 (*Figure 2J*). In addition to *Hand2*, markers for these clusters also included *Alx3, Dlx5,* and *Col2a1*. scRNA-seq analyses further allowed for quantification of which NCC-cell clusters had the most robust Gli3/Hand2 co-expression. We found that 35% of *Gli3+* cells also expressed *Hand2* at E11.5 (*Figure 2—figure supplement 1D*). Furthermore, 50% of *Gli3+* cells in cluster 0,4, and five were also *Hand2+* (*Figure 2—figure supplement 1D*). This was particularly striking since clusters 0, 4 and 5 only accounted for 31% of MNP cells. While cells expressing *Gli* TFs did not organize to any particular clusters, 49% of cells with greater than 1.5 transcripts per million (TPM) *Gli3* expression occupied clusters 0, 4, and 5. Furthermore, 62% of cells with greater than 1.5 TPM *Hand2* expression, also occupied clusters 0, 4, and 5 (*Figure 2—figure supplement 1E*). This was in stark contrast to the other clusters with greater than 1.5 TPM *Hand2* expression (clusters 7, 8, 16), which only account for 14% of the highest expressing *Hand2* cells.

Gene Ontology (GO) analyses for clusters 0, 4, and 5 revealed that these neural crest-derived cells contributed to biological processes altered in *Gli2/3* cKO and *Hand2* cKO mutant embryos, such as skeletal and glossal development, and pattern specification (*Figure 2K*). Additionally, GO-terms for phenotypes arising from dysregulation of these cell clusters included 'decreased tongue size' and 'small mandible' (*Figure 2L*), suggesting that expression of *Gli2/3 and Hand2* in clusters 0, 4, and five may be responsible for the phenotypes present in the conditional knockouts. Since a *Gli2/3* expression gradient or restriction from cell types cannot explain diverse Gli-dependent transcriptional outputs, we hypothesized that functional interactions with Hand2 in clusters 0, 4, and 5 may explain this phenomenon.

## Gli3 and Hand2 occupy CRMs near shared targets in mandibular NCCs

To determine if Gli TFs and Hand2 regulated a common group of target genes, we performed bulk RNA-sequencing on E10.5 *Gli2/3* cKO and *Hand2* cKO MNPs. Transcriptome profiling and GO analyses revealed a wide variety of differentially expressed genes affecting a number of biological processes including 'muscle system process', 'anterior/posterior patterning', 'regionalization', and 'cell-cell signaling' (*Figure 3A–B*). Furthermore, hypergeometric tests revealed significant enrichment of shared transcripts. 50% of genes differentially expressed in *Gli2/3* cKO MNPs were also differentially expressed in *Hand2* cKO MNPs (*Figure 3C*, p=3.7E-284), with 29% being decreased in both mutants and 21% being increased in both mutants (*Figure 3—figure supplement 1A–A'*). This highly significant overlap led us to further investigate mechanisms of a possible co-factorial relationship between Gli TFs and Hand2.

Next, we assessed whether Gli TFs and Hand2 occupied the same CRMs by performing ChIP-seq analyses *in vivo* using endogenously FLAG-tagged alleles for each TF (*Lopez-Rios et al., 2014*; *Osterwalder et al., 2014*; *Lorberbaum et al., 2016*; *Figure 3D*). Since our previous biochemical and expression data supported a unique relationship between Gli3 and Hand2 in the MNP, we focused our characterization of genomic binding on Gli3. As expected, the most highly enriched TF binding site observed in Gli3 ChIP-seq on either E11.5 whole face (frontonasal, maxillary and mandibular prominences) or MNPs alone reflected the previously reported 'canonical' Gli-binding motif (cGBM) defined by the GACCACCC 8-mer (*Kinzler and Vogelstein, 1990*; *Vokes et al., 2008*; *Figure 3E*). Similarly, Hand2 peaks contained both canonical bHLH E-box motifs (CANNTG) and Hand-specific E-box motifs (*Maves et al., 2009*; *Kulakovskiy et al., 2013*; *Figure 3E*). Further motif enrichment analyses revealed that bHLH motifs were also significantly enriched within Gli3 MNP peaks (*Figure 3F*). Comparison between Gli3 and Hand2 MNP ChIP-seq peaks via regulatory

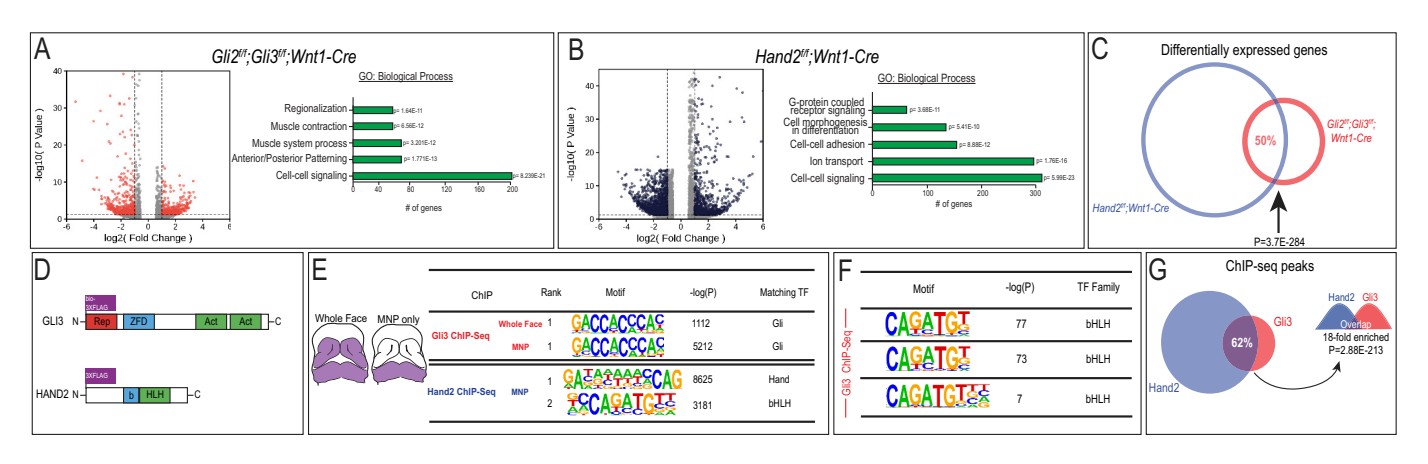

**Figure 3.** Gli3 and Hand2 occupy CRMs near shared targets in the developing MNP. (**A–B**) Volcano plots and GO terms associated with differentially expressed genes from *Gli2^f/f^;Gli3^f/f^;Wnt1-Cre* or *Hand2^f/f^;Wnt1-Cre* E10.5 MNPs (fold change >1.5, adjusted p-value<0.05). (**C**) Venn diagram of shared differentially expressed genes in *Gli2^f/f^;Gli3^f/f^;Wnt1-Cre* and *Hand2^f/f^;Wnt1-Cre* MNPs. (**D**) Endogenously FLAG-tagged mice used for *in vivo* ChIP-seq. (**E**) Known motif enrichment of Gli3 and Hand2 ChIP-seq peaks. (**F**) E-box motif enrichment by HOMER in Gli3 MNP ChIP-seq peaks. (**G**) Venn diagram comparing overlap between Gli3 and Hand2 ChIP-seq peaks, p-value calculated using RELI.

The online version of this article includes the following figure supplement(s) for figure 3:

**Figure supplement 1.** Differential expression in cKO mutants by direction.

element locus intersection (RELI) (*Harley et al., 2018*) revealed a significant overlap of genomic locations occupied by Gli3 and Hand2 in the MNP (*Figure 3G*, 62%, 18-fold enriched, p=2.88E-213).

To determine if the overlap of Gli3 and Hand2 binding at CRMs was biologically relevant, we examined GO-terms associated with genes that were differentially expressed in *Gli2/3* cKO mutants near either Gli3 alone or Gli3/Hand2 overlapping peaks (see Methods). Overall, the GO-terms for differentially expressed genes near Gli3 alone peaks were substantially different from the GO-terms for differentially expressed genes near Gli3/Hand2 overlapping peaks (*Figure 4A*). Interestingly, while GO-terms for differentially expressed genes near Gli3 alone peaks included pattern specification, embryonic organ development and Hh signaling, those associated with differentially expressed genes near Gli3-Hand2 overlapping peaks included a different set of tissue-specific processes including regulation of chondrocyte differentiation and muscle cell differentiation (*Figure 4A*). Not only did it appear that Gli3/Hand2 input conveyed distinct biological relevance, but the number of instances in which differentially expressed genes in *Gli2/3* cKO mutants were near a Gli3/Hand2 overlapping peak were greater than those near a Gli3 peak alone. While approximately 337 differentially expressed genes were associated with a Gli3 alone peak, 463 differentially expressed genes were associated with a Gli3/Hand2 overlapping peak (*Figure 4B*). Finally, to assess how Gli3/Hand2 interactions could be influencing differentially expressed genes, we analyzed the direction of fold change. 17% of genes with a Gli3/Hand2 overlapping peak that were increased in the *Gli2/3* cKO, were also increased in the *Hand2* cKO. Conversely, 33% of genes with a Gli3/Hand2 overlapping peak that were decreased in the *Gli2/3* cKO, were also decreased in the *Hand2* cKO (*Figure 4—figure supplement 1A*). Together, these data suggested that Gli3 and Hand2 were cooperating to positively regulate genes important for mandibular development.

We next superimposed E11.5 single-cell cluster markers onto these findings to reveal that marker genes for NCC clusters had a greater association with Gli3/Hand2 overlapping peaks than Gli3 peaks alone (*Figure 4—figure supplement 1B*). More specifically, genes that were differentially expressed in both *Gli2/3* and *Hand2* cKOs were more likely to be marker genes for clusters 0, 4 and 5 than marker genes for other NCC clusters or other non-NCC clusters (*Figure 4—figure supplement 1C*). To confirm the statistical significance of this finding, we used RELI to test if there was enrichment for scRNA-seq cluster marker genes near Gli3 alone or Gli3/Hand2 overlapping peaks. While there was significant enrichment of cluster marker genes associated with the entire E11.5 MNP near Gli3 alone peaks, there was not a significant enrichment for marker genes for clusters 0,

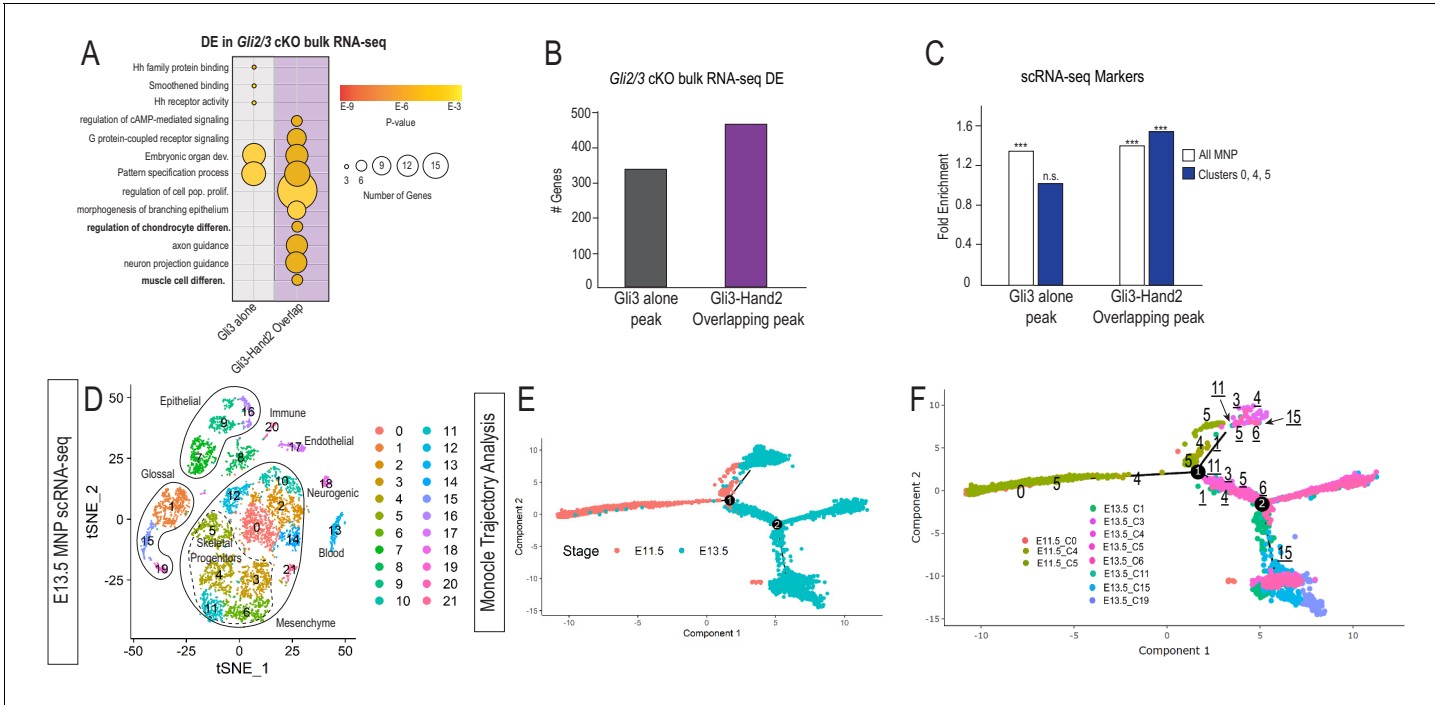

**Figure 4.** Hand2 and Gli3 coordinate glossal and skeletal gene regulatory networks. (A) GO-terms associated with significantly decreased differentially expressed (DE) genes from *Gli2/3* cKO MNP bulk RNA-seq near Gli3 ChIP-seq peaks without Hand2 (Gli3 alone) or near Gli3-Hand2 overlap peaks. (B) Number of DE genes from Gli2/3 cKO MNP bulk RNA-seq near Gli3 ChIP-seq peaks without Hand2 (Gli3 alone) or near Gli3-Hand2 overlap peaks. (C) Enrichment of all MNP clusters or clusters 0, 4, and 5 from E11.5 scRNA-seq near Gli3 ChIP-seq peaks without Hand2 (Gli3 alone) or near Gli3-Hand2 overlap peaks calculated using RELI. ***p<0.001, n.s. not significant. (D) tSNE plot of single-cell RNA-sequencing from E13.5 wild-type MNP. (E–F) Single-Cell Trajectory analysis plot of integrated E11.5 and E13.5 scRNA-seq MNP samples showed the E13.5 glossal (1,15, 19) and skeletal (3, 4, 5, 6, 11) clusters are likely derived from E11.5 Gli3+/Hand2+ NCC clusters (0,4,5). See also *Figure 4—figure supplement 1*.

The online version of this article includes the following figure supplement(s) for figure 4:

**Figure supplement 1.** scRNA-seq and Integration analysis support a subset of NCCs contribute to skeletal and glossal cells of the MNP.

4, 5 (*Figure 4C*). Interestingly, and supportive of our previous data, when we repeated this analysis for Gli3/Hand2 overlapping peaks, we found that there was significant enrichment for cluster marker genes for the entire E11.5 MNP, but also a higher enrichment for marker genes for clusters 0, 4, and 5 (*Figure 4C*). Thus, these analyses suggested a distinct role for the combined action of Gli3 and Hand2 in a subset of NCCs (clusters 0, 4 and 5) during mandibular development.

While our previous data suggested that Gli3/Hand2 interactions conveyed a distinct function in NCC clusters 0, 4 and 5, it was unclear how these clusters contributed to mandibular development. To further delineate the fate of these clusters, we performed scRNA-seq on MNPs at E13.5, a stage when NCC differentiation into distinct cell types had initiated (*Figure 4D*). We used Monocle to perform trajectory analysis on integrated E11.5 and E13.5 scRNA-seq datasets (*Figure 4E,F*; *Figure 4—figure supplement 1D*). These analyses revealed that E11.5 clusters 0, 4 and 5 gave rise to two distinct cell populations at E13.5: the *Myf5 and Myod1* expressing glossal musculature (clusters 1, 15, and 19) and skeletogenic progenitors (clusters 3, 4, 5, 6, and 11), marked by many osteochondrogenic genes including *Sp7, Runx2, Sox9, Col1a1, Col9a2,* and *Barx1*. These findings were consistent when Integration Analysis and re-clustering of these datasets was performed and visualized using UMAP (*Figure 4—figure supplement 1E–F'*). Together, these analyses suggested that Gli3/Hand2 interactions were enriched in E11.5 clusters 0, 4 and 5, which in turn give rise to skeletogenic and glossal components of the lower jaw.

## Low-affinity Gli-binding motifs are within close proximity to E-boxes and specific to the developing mandible

Collectively, our genetic analysis, expression profiling and TF binding data supported a critical role for Gli3/Hand2 interactions during mandibular development; however, the specific mechanisms underlying combinatorial transcriptional regulation for shared Gli3/Hand2 targets was unclear. To further investigate potential co-regulatory interactions, we performed *de novo* motif analysis on Gli3-alone vs. Gli3/Hand2-overlapping peak regions. As expected, the most enriched motif within Gli3-alone peaks was the previously reported 'canonical' GBM (**c**GBM) defined by the 'GACCACCC' 8-mer (*Kinzler and Vogelstein, 1990*), which was 9.8-fold-enriched compared to background sequences (*Figure 5A*). Surprisingly, when we performed motif analysis on overlapping peaks shared between Gli3 whole face and Hand2 MNP samples, the top-ranked GBM (6.3-fold-enriched over background sequences) deviated from the **c**GBM 8-mer, with the most notable change being the reduced weight of the highly conserved 'A' at the 5th position (*Figure 5A'*). To specifically address the Gli3/Hand2 relationship in the MNP, we repeated these analyses using only overlapping peaks from Gli3 and Hand2 MNP samples. Here, the top-ranked GBM present (6.2-fold-enriched over background) differed even further from the canonical 8-mer, having a higher probability of either a 'T' rather than 'A' at the highly constrained 5th position (*Figure 5A''*). We designated this GACC**T**CCC 8-mer as a 'divergent' GBM (**d**GBM). Interestingly, the **d**GBM was most clearly revealed upon comparisons between MNP data sets, with 85% of Gli3/Hand2 overlapping peaks containing **d**GBM and only 9% of Gli3/Hand2 overlapping peaks contained a **c**GBM. 6% of Gli3/Hand2 overlapping peaks contained neither a **c**GBM or a **d**GBM (*Figure 5—figure supplement 1A*). These data supported the possibility that the **d**GBM utilized by Gli3 and Hand2 was specific to the MNP. To test this hypothesis, we repeated our *de novo* motif analysis comparing to publicly available data from the developing limb (*Figure 5B*; *Osterwalder et al., 2014*). Strikingly, the **d**GBM present in our MNP analysis was not present when comparing Hand2 binding in the limb. Rather, these analyses revealed the highly constrained 5th position remained exclusively a heavily weighted 'A'. Together, these data suggested a tissue-specific role for this MNP-enriched **d**GBM.

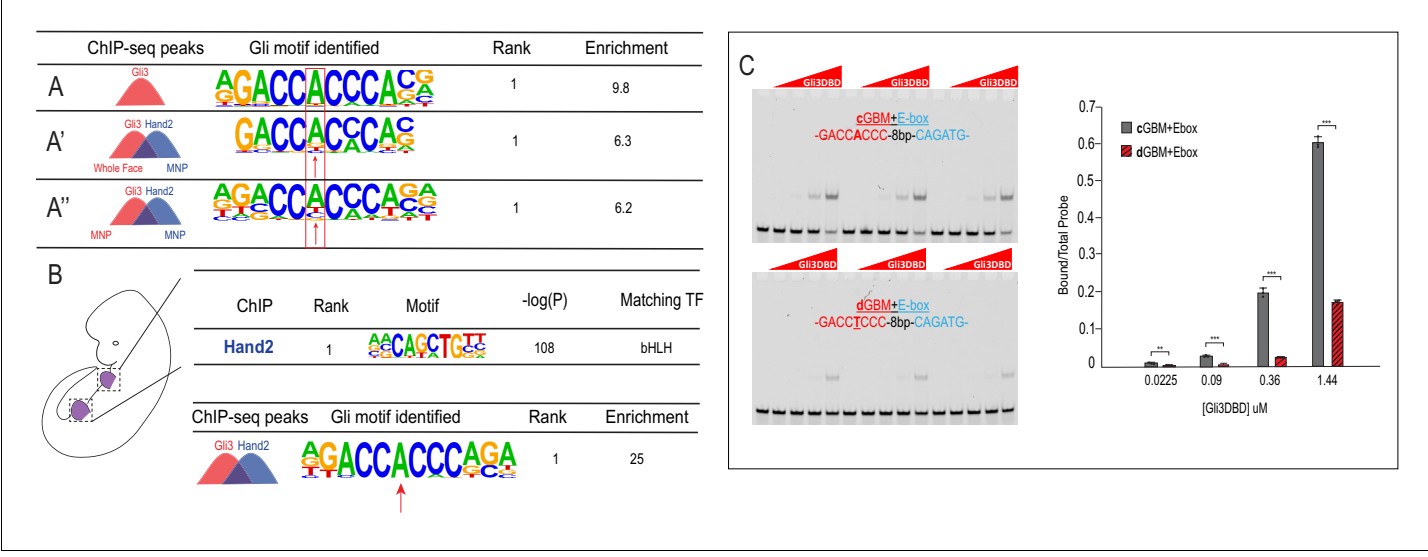

**Figure 5.** Low-affinity divergent Gli-binding motifs are found near E-boxes. (A–A'') *De novo* motif enrichment for Gli3-only peaks in MNP, Gli3/Hand2 overlapping peaks, comparing (A') Gli3-whole face peaks to Hand2 MNP peaks or (A'') Gli3 MNP peaks to Hand2 MNP peaks. (B) (Top) Known motif enrichment of Hand2 peaks from limb buds of endogenously FLAG-tagged mice. (Bottom) *De novo* motif enrichment of Gli3/Hand2 overlapping peaks, comparing Gli3 peaks from whole face and Hand2 peaks from limb. (C) Electrophoretic mobility shift assay (EMSA) and quantification of affinity showing that the Gli3 DNA-binding domain (Gli3DBD) binds with increased affinity to canonical GBMs (**c**GBM) relative to divergent GBMs (**d**GBMs). Results used for quantification are shown in triplicate, *p<0.05, **p<0.01, ***p<0.001. See also *Figure 5—figure supplement 1*.

The online version of this article includes the following source data and figure supplement(s) for figure 5:

**Source data 1.** Results from Simple counting method of quantifying instances of GBMs in ChIP-seq data.
**Figure supplement 1.** Gli3-Hand2 overlapping ChIP-seq peaks that contain a cGBM or dGBM.

To confirm the decreased frequency of **c**GBM binding events in the presence of Hand2 in the MNP, we quantified the incidence of the **c**GBM 8-mers using a strict counting method. While the consensus **c**GBM 8-mer (GACCACCC) was detected in 16% of Gli3-only peaks collected from the MNP, its occurrence was significantly reduced to only 2% of Gli3/Hand2 overlapping peaks. Additionally, while the **c**GBM was the 33[rd] most frequent 8-mer in Gli3-only MNP peaks (out of 32,896 possibilities), it was 563[rd] in frequency in Gli3/Hand2 overlapping MNP peaks (*Supplementary file 2*). This finding, in conjunction with the motif enrichment results, further supported a deviation from the **c**GBM when Hand2 and Gli3 peaks overlapped.

To examine the effect an 'A' to 'T' transition at the 5[th] position had on relative binding affinity, we utilized previously published Gli3 protein-binding microarray (PBM) E-score data (*Peterson et al., 2012*). PBM E-scores range from −0.5 to +0.5, with values above 0.4 generally considered strong binding sites (*Berger et al., 2006*; *Berger and Bulyk, 2009*). Interestingly, substitution of 'A' to 'T' in the 5[th] position of comparable 8-mers reduced the E-score for Gli3 binding from 0.42 to 0.33, indicating that Gli3 has a lower affinity for the **d**GBM sequence. Likewise, previous studies in *Drosophila* reported that low-affinity non-canonical GBMs with a 'T' in the 5[th] position, similar to what we term the **d**GBM, were responsible for regulating broad expression of Ci targets in zones of lower Hh signaling (*Parker et al., 2011*). To directly test the binding affinity of Gli3 to a **d**GBM, we performed gel-shift assays on synthetic sequences containing either a **d**GBM+E-box or **c**GBM+E-box. These experiments confirmed that a single nucleotide alteration from 'A' to 'T' in the 5[th] position significantly decreased the affinity of Gli3 DNA binding (*Figure 5C*). Together, these data confirmed the identification of distinct, low-affinity **d**GBMs enriched at genomic loci bound by both Gli3 and Hand2 within the MNP.

## dGBMs direct unique gene regulatory programs in neural crest-derived skeletal and glossal progenitors of the MNP

To specifically address the possible functional consequences of utilization of a **c**GBM vs. **d**GBM, we superimposed our motif analysis on GO-terms associated with genes that were differentially expressed in *Gli2/3* cKO mutants near Gli3/Hand2 overlapping peaks (*Figure 6A*). Overall, the GO-terms associated with **c**GBMs were substantially different from those associated with **d**GBMs. Furthermore, GO-terms associated specifically with the **d**GBM included a muscle-specific subset. We next examined the prevalence of **d**GBMs near genes that were differentially expressed in conditional KO mutants and near Gli3/Hand2 overlapping peaks. These analyses revealed that relatively few differentially expressed genes were associated with peaks containing a **c**GBM, whereas many more differentially expressed genes were associated with peaks containing a **d**GBM (*Figure 6B*).

To determine if these trends hold true in NCCs specifically, we combined our motif analyses with GO-terms enrichment analysis for NCC cluster marker genes near Gli3-Hand2 overlapping peaks. These analyses suggested that **c**GBMs were significantly associated with neurogenic biological processes, whereas **d**GBMs were specifically associated with skeletogenic biological processes (*Figure 6C*). We next quantified the percentage of NCC cluster marker genes near **c**GBMs and **d**GBMs. Using RELI, we determined that there was significant enrichment of overlapping peaks containing **d**GBMs in NCC clusters (*Figure 6D*). Furthermore, overlapping peaks containing a **d**GBMs were enriched near NCC cluster marker genes, when compared to all other clusters (*Figure 6E*). Together, these analyses suggested that the presence of a **d**GBM was associated with genes differentially expressed in *Gli2/3* and *Hand2* conditional mutants and that it was specifically enriched in NCCs that give rise to skeletal and glossal derivatives.

Having identified a global trend of **d**GBM association with NCC marker genes, we sought to identify specific targets from our transcriptome and ChIP-seq analyses relevant for MNP development. We chose four targets relevant to MNP development including *Forkhead Box d1* (*Foxd1*) a well-characterized Gli target involved in MNP regionalization (*Jeong et al., 2004*); *Pleiomorphic adenoma gene-like 1* (*Plagl1*), a gene which impacts glossal development (*Li et al., 2014*); *Myosin heavy chain 6* (*Myh6*), a myosin isoform found in specialized skeletal muscles (*Lee et al., 2019*) and *Avian Musculoaponeurotic fibrosarcoma oncogene homolog* (*Maf*), a TF involved in chondrocyte differentiation (*Hong et al., 2011*). To identify regions with potential regulatory function, we integrated Cis-BP-identified (*Weirauch et al., 2014*) **c**GBMs with our MNP-specific ATAC-seq and ChIP-seq data to highlight regions of open chromatin that were bound by Gli3 and Hand2. Interestingly, we frequently saw areas of open chromatin occupied by Gli3 and Hand2 that did not contain a high-affinity **c**GBM

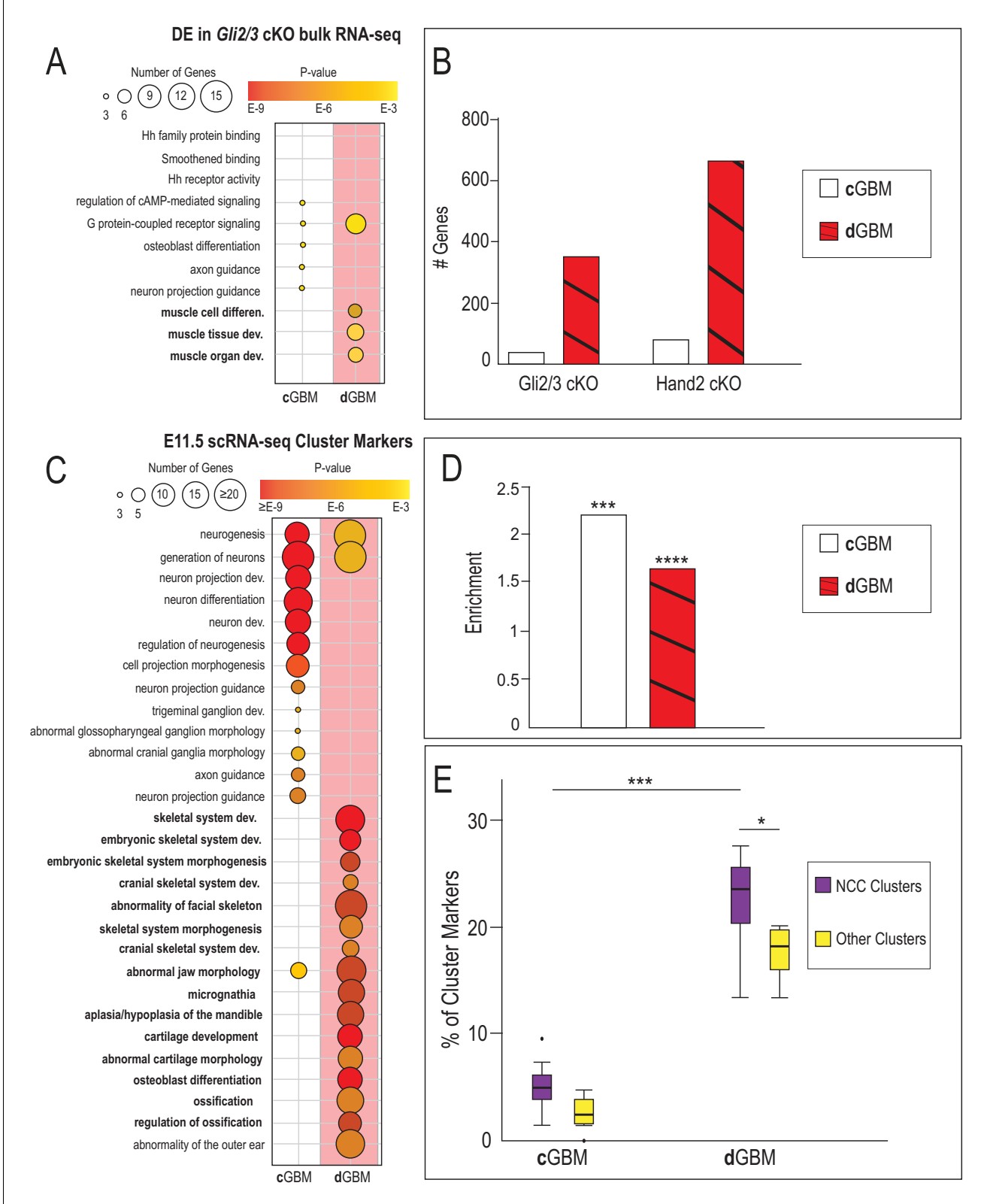

**Figure 6.** Divergent GBMs direct unique GRNs in neural crest-derived skeletal and glossal progenitors of the MNP. (**A**) GO-terms associated with significantly decreased differentially expressed (DE) genes from *Gli2/3* cKO MNP bulk RNA-seq near Gli3-Hand2 overlap peaks with a **c**GBM and with a **d**GBM. (**B**) Number of DE genes from *Gli2/3* cKO or *Hand2* cKO MNP bulk RNA-seq near Gli3-Hand2 overlap peaks with a **c**GBM and with a **d**GBM, *p<0.05, **p<0.01, ***p<0.001, ****p<1E-11. (**C**) GO-terms associated with E11.5 scRNA-seq NCC clusters near near Gli3-Hand2 overlap peaks with a

*Figure 6 continued on next page*

Figure 6 continued

cGBM and with a dGBM reveal distinct mechanistic consequences when a dGBM is present in Gli3-Hand2 overlap peaks. (D) Enrichment of E11.5 scRNA-seq NCC cluster markers in genes near Gli3-Hand2 overlap peaks with a cGBM and with a dGBM using RELI. (E) Box and whisker plots showing percent of E11.5 scRNA-seq cluster markers near Gli3-Hand2 overlap peaks with a cGBM and with a dGBM. Significantly higher overlap in NCC cluster markers is seen near Gli3-Hand2 overlap peaks with a dGBM compared to those with a cGBM.

(*Figure 7A–D*, black lines). Instead, these loci all displayed Gli3 and Hand2-bound regions containing dGBMs (*Figure 7A–D*, red lines). This was in stark contrast to the Gli3-bound areas of open chromatin heavily populated with cGBMs at the *Ptch1* locus (*Figure 7—figure supplement 1A*, black lines). Furthermore, all four of the selected target genes were initially expressed in E11.5 clusters 0,

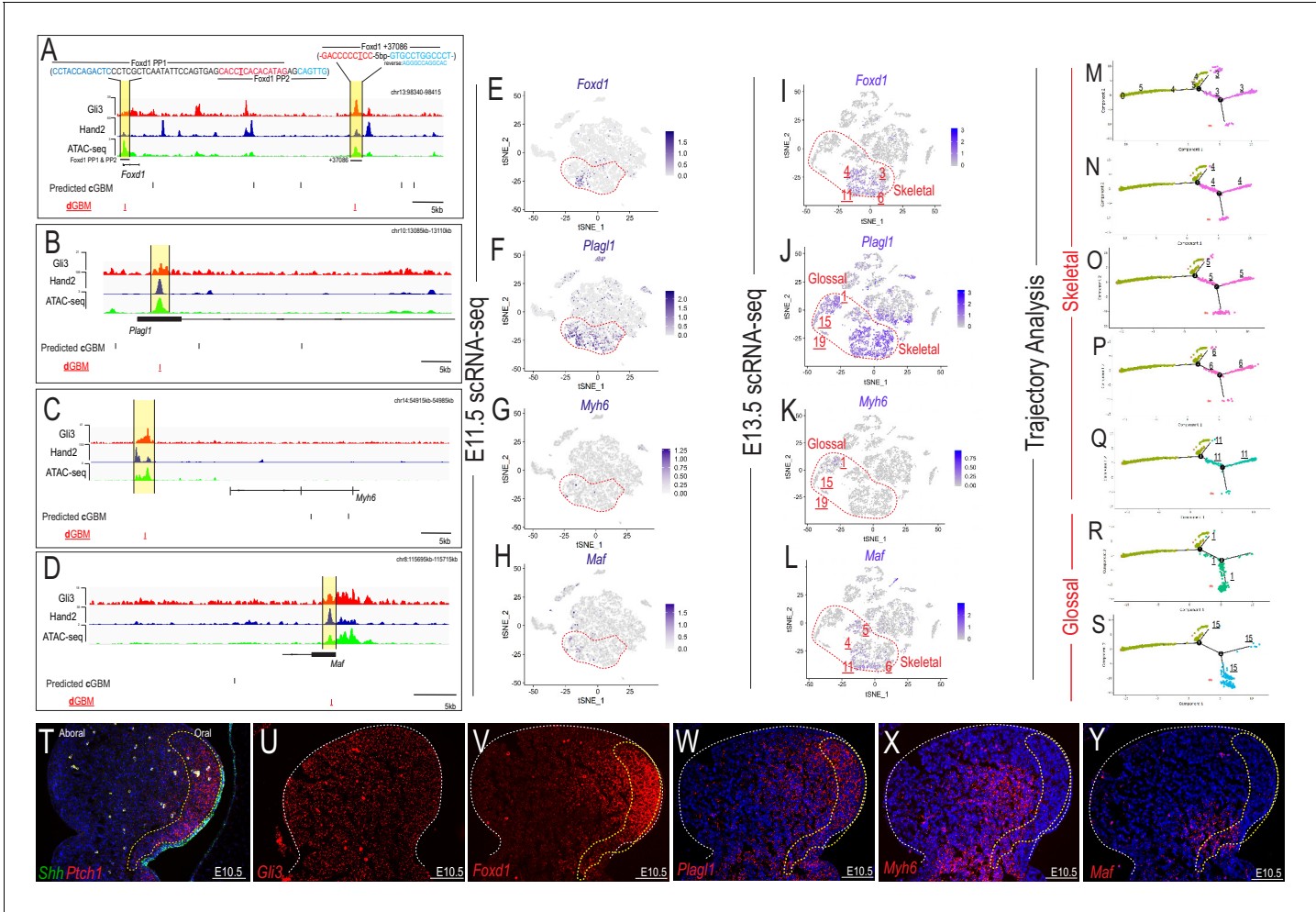

**Figure 7.** Hand2 correlates with non-canonical Gli-responsive expression patterns. (A–D) Overview of MNP-specific regulatory input to the *Foxd1, Plagl1, Myh6,* and *Maf* locus. cGBMs (black line) and dGBMs (red lines) are indicated below the signal tracks for Gli3 (red) and Hand2 (blue) ChIP-seq and ATAC-seq (green). PP1 = promoter proximal 1, PP2 = promoter proximal 2. (E–H) Single-cell expression of *Foxd1, Plagl1, Myh6,* and *Maf* in the E11.5 MNP. Dotted red line indicates *Gli+/Hand2+* NCC clusters (0, 4, 5). (I–L) E13.5 scRNA-seq expression in *Gli3/Hand2+* -derived clusters of Gli3 and Hand2 targets involved with MNP patterning. (M–S) Single-cell Trajectory analysis plot of integrated E11.5 and E13.5 scRNA-seq MNP samples highlighting E11.5 clusters 0, 4, and 5 likely give rise to the E13.5 glossal and skeletal clusters (T–U) Expression of *Shh, Ptch1, and Gli3* as revealed by smFISH in sagittal sections of E10.5 MNPs. Dotted yellow line indicates highest Shh-responsive area marked by *Ptch1*. (V–Y) smFISH expression of Gli3 and Hand2 targets involved with MNP patterning. Scale bar: 100 µm. See also *Figure 7—figure supplement 1*.

The online version of this article includes the following figure supplement(s) for figure 7:

**Figure supplement 1.** *Ptch1* is activated in response to high Shh through canonical GBMs.

4, and 5 (*Figure 7E–H*), which were shown to give rise to NCC-derived skeletal and glossal derivatives in the E13.5 MNP (*Figure 7I–L*), via trajectory analysis (*Figure 7M–S*).

To follow up on differences in GBM quality/variants observed between *Ptch1* and our identified target genes, we examined expression patterns for all four genes in E10.5 MNPs using RNAscope. As expected, *Ptch1* was expressed in neural crest mesenchyme directly adjacent to an epithelial source of *Shh* on the oral axis of the MNP and was indicative of a high level of Shh pathway activity (*Figure 7T*, dotted yellow line). As previously described, *Gli3* expression was uniformly observed throughout the oral-aboral axis of the MNP (*Figure 7U*). Interestingly, all four of our identified target genes (*Foxd1, Plagl1, Myh6* and *Maf*) were expressed both within and outside of the *Ptch1* domain in neural crest-derived mesenchyme of the MNP (*Figure 7T–Y*). To quantify this observed phenomenon, we used our E11.5 scRNA-seq to determine that the majority of *Foxd1+, Plagl1+, Myh6+,* or *Maf+* NCCs did not express *Ptch1* (*Figure 7—figure supplement 1B*). Thus, our data suggested that Hand2 and Gli3 collaborate at **d**GBMs to activate transcriptional networks within MNP NCCs to establish osteogenic, chondrogenic, and glossal/muscle cell fates. These data further suggested that despite being Gli3 targets, these genes did not require graded Shh activity for expression, but rather utilized combined input from Gli3/Hand2.

## Gli3 and Hand2 synergize at dGBMs

Cooperating TFs frequently bind with a preferred spacing and orientation (*Jolma et al., 2013*; *Narasimhan et al., 2015*). To further understand the mechanisms of Gli3 and Hand2 cooperation at CRMs containing **d**GBMs, we tested if there was a statistical preference for any single spacing or orientation of GBMs and E-boxes inside of Gli3/Hand2 overlapping genomic regions using the previously published COSMO method (*Narasimhan et al., 2015*). Despite identifying 628 'intersecting peaks' that contain a GBM and Hand2 motif within 100 bases of each other, no particular spacing/configuration was present in greater than ~0.2% of sequences (*Figure 8—figure supplement 1A*; *Supplementary file 3*). These findings suggested flexibility in the regulatory architecture governing the spacing and orientation of Gli3 and Hand2 binding sites within CRMs bound in the developing MNP. Based on these results, we identified three potential regulatory regions near the *Foxd1* promoter for further functional analysis. We designated a region containing a **d**GBM 22 base pairs downstream of an E-box and two base pairs upstream of a second E-box as promoter proximal 1 (PP1) and promoter proximal 2 (PP2), respectively, and designated a second putative regulatory region downstream of the *Foxd1* coding region as +37086 (*Figures 7A* and *8A*).

While ChIP data was highly suggestive of Gli3/Hand2 co-occupancy at regulatory regions containing **d**GBM and E-box motifs, it did not test if Gli3 and Hand2 were able to simultaneously bind an endogenous **d**GBM and an adjacent E-box. To address this question, we performed gel-shift assays with the Gli3 DNA-binding domain and full-length Hand2 (Hand2FL) on putative endogenous CRMs near the *Foxd1* locus. Gel-shift analysis revealed that the Gli3 DNA-binding domain independently binds the **d**GBM present in PP1 and PP2 (*Figure 8A*). In addition, we found that Hand2 could not independently bind the E-box motifs present in the PP1 and PP2 probes but could bind as a heterodimer in the presence of E47L (Tcf3), an E-protein bHLH that cooperatively binds DNA with many tissue-specific bHLHs (*Figure 8A*). This is consistent with reports that binding of many bHLH TFs require dimerization with other widely expressed E-protein family members (*Firulli, 2003*). Importantly, Gli3 and Hand2 were able to simultaneously bind **d**GBM/E-box regions within both PP1 and PP2 (*Figure 8A*), in a dose-dependent manner (*Figure 8—figure supplement 1B*). Together, these data suggested that Gli3 and Hand2 simultaneously occupy potential regulatory regions containing a low-affinity **d**GBM and an E-box. We next sought to investigate how Gli3/Hand2 cooperation impacted transcriptional output.

To examine Gli3/Hand2 transcriptional activity, we generated luciferase reporter constructs containing either PP1, PP2 or +37086 putative *Foxd1* regulatory regions regulatory regions that contained **d**GBMs and E-boxes. Constructs were transfected into the cranial NCC line, O9-1 (*Ishii et al., 2012*) and luciferase activity was measured after transfection of Gli3 alone, Hand2 alone or Gli3 and Hand2 together (*Figure 8B–D*; *Figure 8—figure supplement 2A*). Gli3 alone induced luciferase expression in *Foxd1* PP1, PP2, and +37086 (*Figure 8B–D*). Hand2 alone induced activity of PP1 and +37086 but did not significantly increase luciferase activity of PP2 relative to the control. Similar to the output observed with synthetic constructs, co-expression of Gli3 and Hand2 elicited significant and synergistic outputs at all three putative regulatory elements containing endogenous **d**GBMs

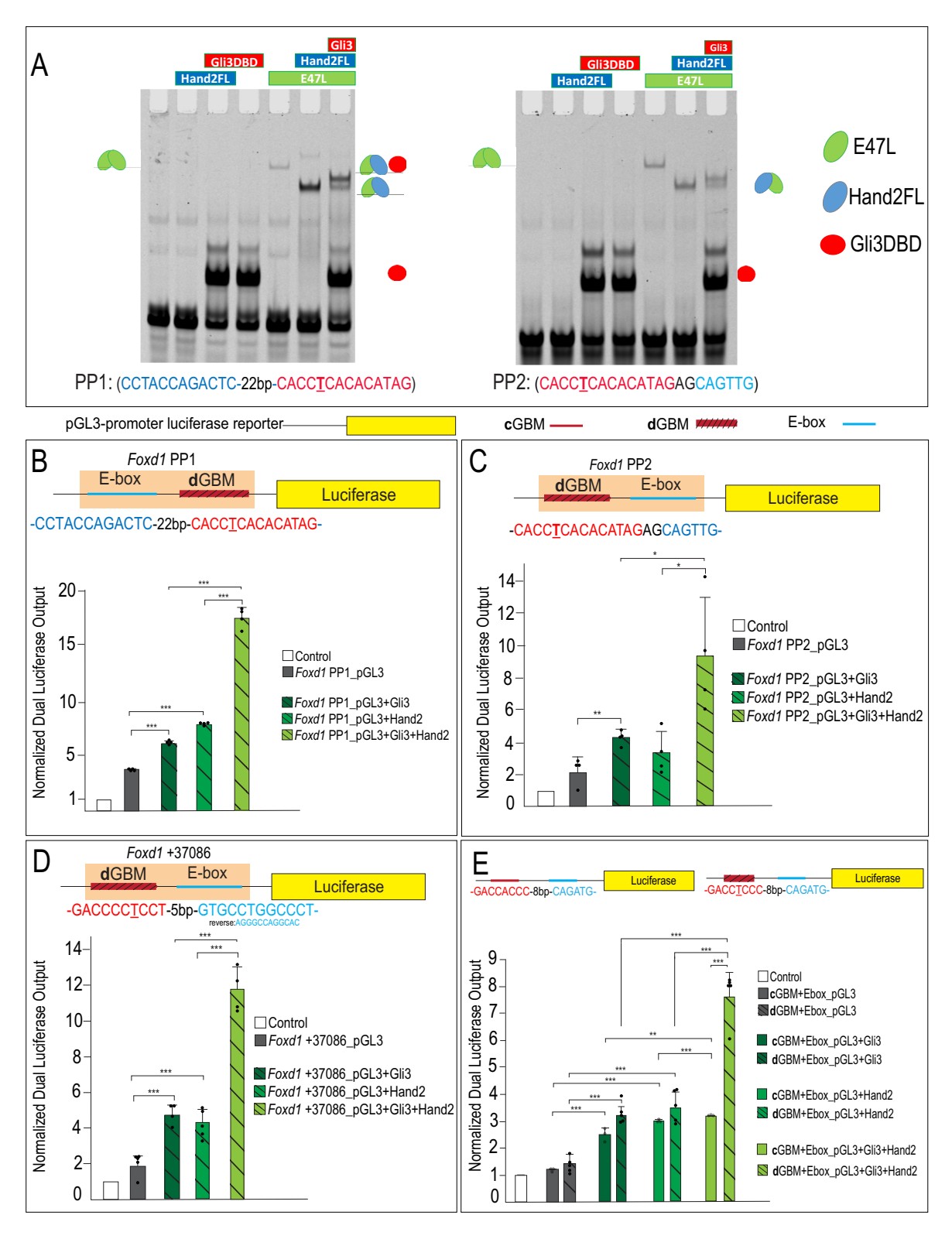

**Figure 8.** Gli3 and Hand2 synergistically activate low-affinity **d**GBMs. (**A–C**) Luciferase reporter activity of the endogenous *Foxd1* putative regulatory region fragments PP1, PP2, and +37086 after transfection with Gli3, Hand2, or both in O9-1 cells. (**D**) Luciferase reporter activity of synthetic constructs containing a **c**GBM and E-box (solid bars) or **d**GBM and E-box (hatched bars) in response to transfection of Gli3, Hand2, or both in O9-1 cells. Data are expressed as mean + SD with biological replicates shown as dots. *p<0.05, **p<0.01, ***p<0.001. See also *Figure 8—figure supplements 1* and *2*.
*Figure 8 continued on next page*

*Figure 8 continued*

The online version of this article includes the following source data and figure supplement(s) for figure 8:

**Source data 1.** Pooled ChIP-seq replicate peak calls.
**Figure supplement 1.** No observed enriched spacing or orientation of GBM and Ebox.
**Figure supplement 2.** Gli3 and Hand2 co-expression synergistically activates *Foxd1 in vitro*.
**Figure supplement 2—source data 1.** Results from COSMO algorithm.

(*Figure 8B–D*; light green hatched bars). This surprising synergism between Gli3 and Hand2 was further confirmed *in vitro* by examining *Foxd1* expression in O9-1 cells, where the presence of Gli3 and Hand2 culminated in synergistic expression of *Foxd1* (*Figure 8—figure supplement 2B*).

To confirm that the observed synergism was dependent upon the presence of Gli3 and Hand2 with a **d**GBM plus an E-box, we generated synthetic luciferase reporter constructs containing either the **c**GBM or a **d**GBM plus an E-box (**c**GBM+E-box, **d**GBM+E-box, respectively) and again transfected O9-1. Luciferase activity was measured after transfection of Gli3 alone, Hand2 alone or Gli3 and Hand2 together (*Figure 8E*). Regardless of the GBM present, expression of either Gli3 alone or Hand2 alone significantly elevated the luciferase activity of reporter constructs relative to control conditions (*Figure 8E*). Co-expression of Gli3 and Hand2 with the **c**GBM+E-box synthetic reporter resulted in a small, but significant increase in luciferase expression compared to either Gli3 or Hand2 alone. In stark contrast, co-expression of Gli3 and Hand2 in the presence of the **d**GBM+E-box synthetic reporter resulted in a significant and synergistic (more than additive) upregulation of luciferase expression compared to either Gli3 or Hand2 alone (*Figure 8E*). Together, these results indicated that (1) the low-affinity **d**GBM conveyed a distinct function from the **c**GBM, (2) the low-affinity **d**GBM+E-box produced synergistic transcriptional output in the presence of Gli3 and Hand2 and (3) synergistic activity was independent of a graded Hh signal, since the response was observed without Hh stimulation.

To confirm that this synergism was dependent upon the presence of both a **d**GBM and E-box, we performed site-directed mutagenesis. Mutation of either the **d**GBM or E-box sequence eliminated synergistic output in *Foxd1* endogenous putative regulatory regions (*Figure 9A*; *Figure 9—figure supplement 1A*). Furthermore, to determine if the central 'T' which we used to define **d**GBMs was causative for the synergistic output, we mutated the 'T' in the PP2 putative regulatory region to an 'A', resembling a **c**GBM. This single base-pair 'T > A' change significantly increased affinity of Gli3 for the GBM and abolished the synergistic luciferase output when Gli3 and Hand2 were co-expressed (*Figure 9B–E*). Together, these data support a novel, tissue-specific transcriptional mechanism in which Gli3 and Hand2 utilize low-affinity **d**GBM and E-boxes to promote synergistic activation of *Foxd1* (and likely other MNP targets) outside of a Hh gradient (*Figure 10*).

## Discussion

Substantial evidence has long supported the idea that the Hh signaling pathway utilizes a morphogen gradient to convey a threshold of activation responses necessary to pattern tissues throughout the embryo (*Dessaud et al., 2008*). While the concept of a morphogen gradient has been supported by several biochemical and genetic studies, a significant gap remains in understanding the mechanisms of how cells perceive and transduce morphogens. This knowledge gap is especially evident within the developing craniofacial complex, where despite requiring a localized, epithelial Hh source, neither a *Gli* gradient nor a primary requirement of a single Gli (e.g. Gli2 or Gli3) is apparent within facial prominences. In this study, we have uncovered a unique mechanism used in the developing mandible that produces synergistic target gene responses outside of a traditional morphogen gradient by utilizing regulatory elements containing low-affinity GBMs that integrate input from a tissue-specific binding partner. Specifically, our results establish a novel relationship between Gli3 and Hand2, in which these factors synergize at low-affinity 'divergent' GBMs (**d**GBMs) for a subset of target genes important for key processes in mandibular development including patterning, skeletogenesis and glossogenesis (*Figure 10*). To our knowledge, this is the first study to identify and explore variable levels of Gli-dependent transcriptional activity across a field of cells as a mechanism for generating cellular identities in the developing face.

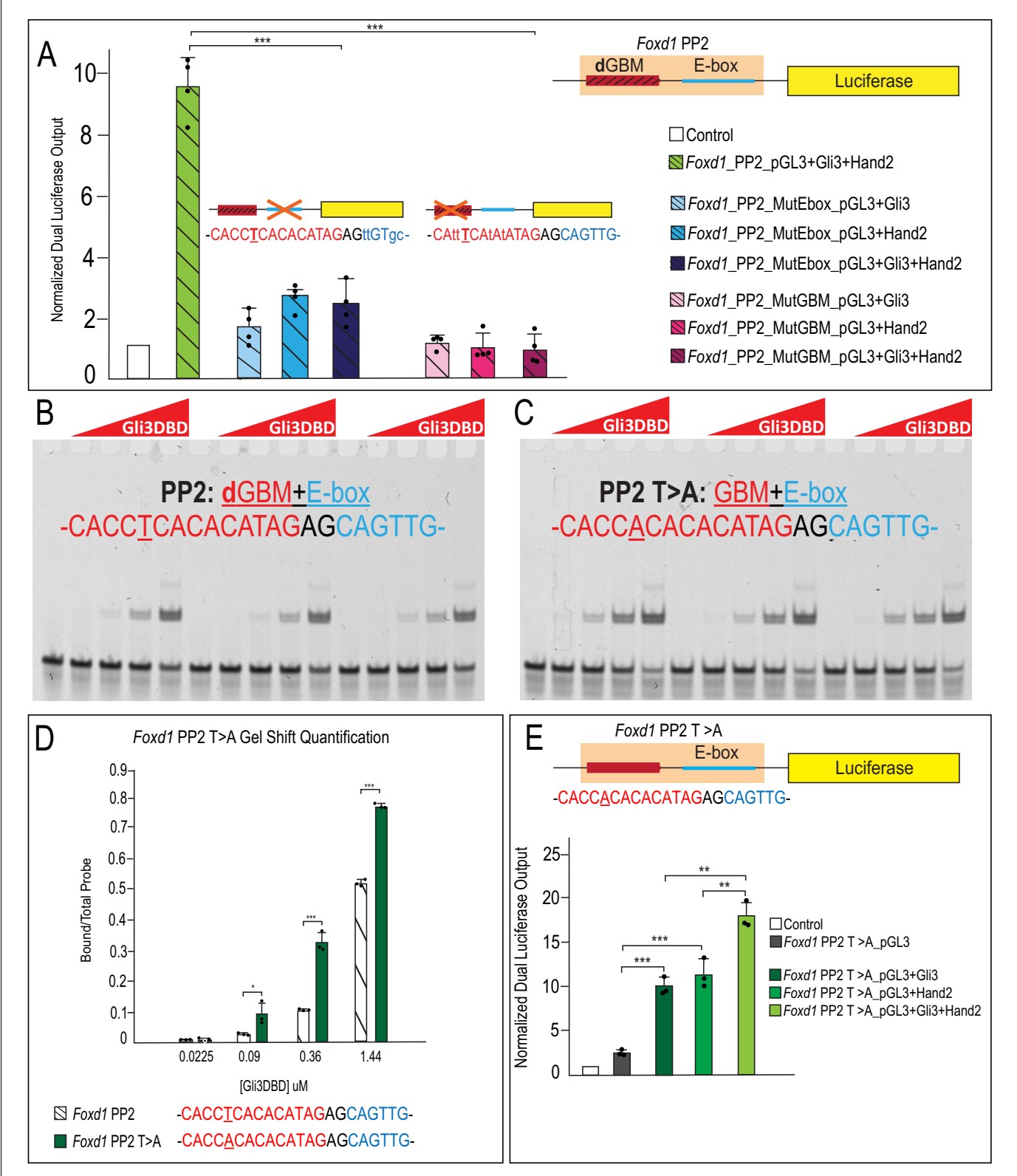

**Figure 9.** Occupancy of low-affinity **d**GBM and E-box are required for synergism. (**A**) Luciferase reporter activity of mutant GBM or E-box motifs from *Foxd1* PP2 showing mutation of E-box or GBM abolishes synergistic activation. (**B–C**) EMSA for Gli3DBD binding affinity of (**C**) endogenous *Foxd1* PP2 or (**D**) T > A mutant *Foxd1* PP2. (**D**) Quantification of (**B**) and (**C**) showing increased Gli3DBD binding affinity of endogenous *Foxd1* PP2 (white hatched) compared to T > A mutant *Foxd1* PP2 (green). (**E**) Increased luciferase reporter activity when T > A change is made within *Foxd1* PP2. Data are

*Figure 9 continued on next page*

*Figure 9 continued*

expressed as mean + SD. Luciferase data have biologic replicates shown as dots. *p<0.05, **p<0.01, ***p<0.001. See also *Figure 9—figure supplement 1*.

The online version of this article includes the following source data and figure supplement(s) for figure 9:

**Source data 1.** GO terms associated with Differentially Expressed Genes.
**Figure supplement 1.** Mutations of **d**GBM or E-box of *Foxd1* +37086 putative enhancer abolish Gli3-Hand2 synergism.
**Figure supplement 1—source data 1.** hared ChIP-seq peaks between replicates.

## Low-affinity GBMs function as important transcriptional determinants

TF binding site affinity is one mechanism utilized by cells in other tissues to produce graded threshold responses (*Driever et al., 1989*; *Oosterveen et al., 2012*; *Peterson et al., 2012*). The established model states that target genes within a high concentration of the morphogen gradient are activated through low-affinity sites (*Jiang and Levine, 1993*), whereas those exposed to lower morphogen concentrations utilize high-affinity sites (*Ip et al., 1992*). Despite the validation of this idea in many contexts, regulation of several Hh targets are inconsistent with this model. For example, in the *Drosophila* imaginal disc, *ptc* is restricted to the highest Hh threshold and is regulated by high-affinity canonical GBMs, whereas, *dpp* is expressed broadly throughout the Hh gradient and is regulated by low-affinity non-canonical GBMs (*Wang and Holmgren, 1999*; *Parker et al., 2011*). Previous ChIP studies in the developing limb have reported that while 55% of Gli-binding regions contained a high-affinity GBM, the remaining 45% of regions contained a low-affinity GBM or no GBM. Interestingly, low-affinity GBMs are strongly conserved across both tissues and species (*Vokes et al., 2008*; *Parker et al., 2011*). Furthermore, the same study also reported that a small number of Gli-binding regions contained limb-specific variants of the GBM, supporting previous

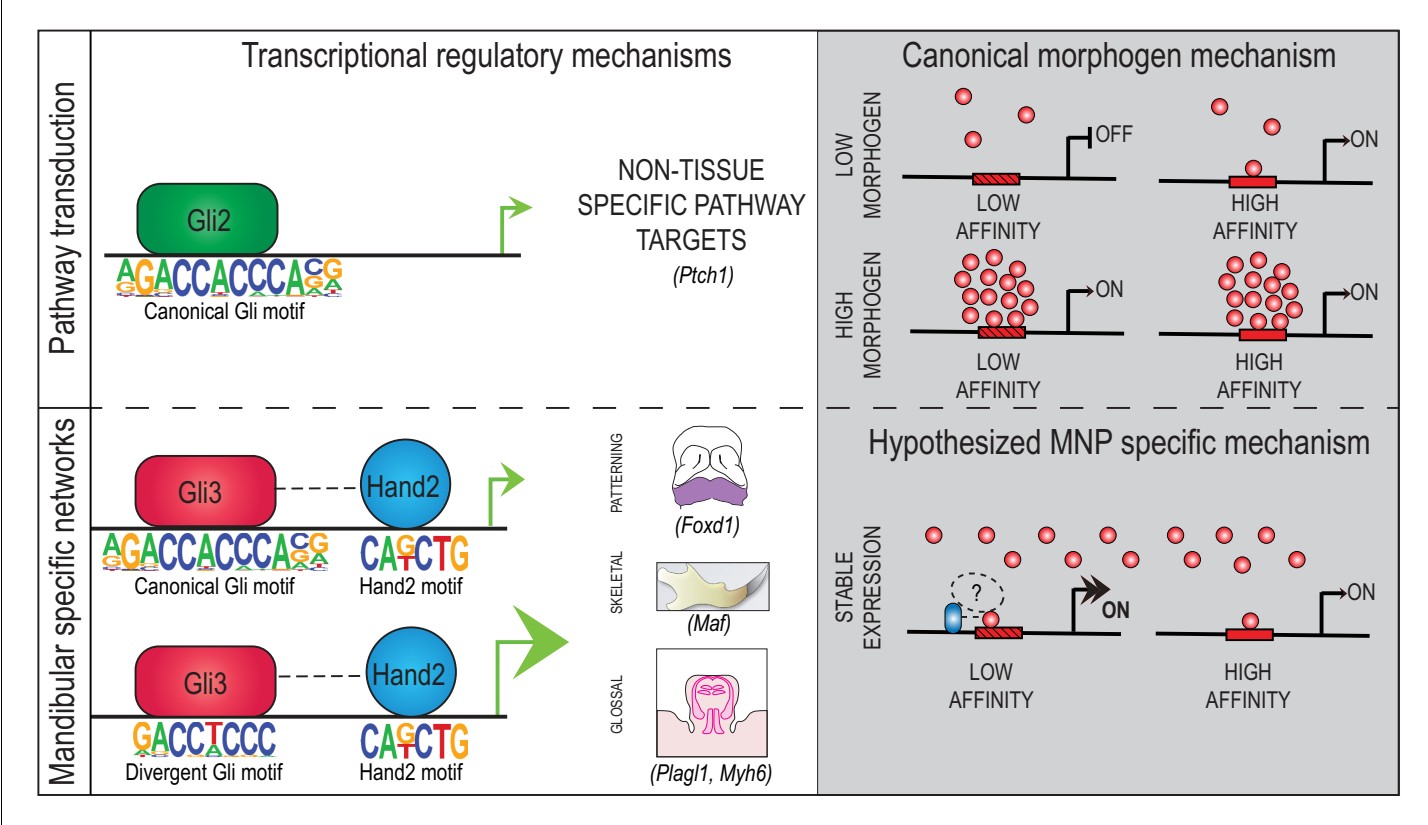

**Figure 10.** Model of Gli3-Hand2 cooperation in the developing MNP. Model of Gli3-Hand2-specific cooperation at low-affinity **d**GBMs drives patterning, skeletal and glossal GRN in regions of the MNP outside of the highest Shh ligand concentration.

reports that low-affinity motifs are absolutely critical to confer spatially distinct gene expression (*Jiang and Levine, 1993*; *Lebrecht et al., 2005*; *Vokes et al., 2008*). Our findings are the first to report how GBM affinity is utilized in a craniofacial context. In the face, we identified low-affinity, divergent binding sites that were necessary and sufficient to drive robust gene expression required for mandibular development (*Figure 10*). Interestingly, as no discernable concentration gradient of *Gli2/Gli3* in the developing mandible exists, the utilization of these low- affinity divergent sites is likely not dictated by a graded Hh signal. It should be noted, that while Gli3 is capable of binding to **d**GBMs on its own, motif enrichment analyses did not reflect this occurring at a high frequency *in vivo*. Thus, these data suggested that co-factors such as Hand2 may be necessary to 'recruit' Gli3 to **d**GBMs in proximity to an E-box. This hypothesis is further supported by the fact that despite *Hand2* having a more restricted expression domain than *Gli3* within the developing MNP, conditional loss of *Hand2* alone in NCCs generates a more severe mandibular phenotype than that observed in *Gli2/Gli3* conditional mutants. Additional studies will be necessary to fully understand the complex network of inputs that contribute to GBM binding specificity.

Previous studies examining Gli-binding in the limb and central nervous system (CNS) have identified E-boxes within Gli ChIP-seq peaks. *De novo* motif analysis revealed an E-box enriched in limb Gli-binding regions with or without a high-affinity GBM (*Vokes et al., 2008*). At the time, the significance of the E-box to Gli3 transcriptional activity was unknown. In the developing CNS, an E-box was the second ranked motif identified in Gli1 ChIP-seq peaks (*Lee et al., 2010*). Mutational analyses determined these E-boxes had varying (context-specific) effects on Gli-mediated transcription, sometimes conferring no affect, while in other cases reducing Gli1-responsiveness (*Lee et al., 2010*). Our studies significantly advance these findings by demonstrating that co-utilization of GBMs and E-boxes allows Gli TFs to utilize lower affinity sites and produce synergistic transcriptional outputs. Furthermore, our mutational analyses revealed that a single base-pair substitution ('A' with a 'T' at the central 5[th] residue) was sufficient to convey both affinity and synergism. Interestingly, similar divergent, low-affinity GBMs with a medial 'T' were previously reported in *Drosophila* within the *dpp* enhancer (*Parker et al., 2011*). In light of the *dpp* expression pattern, which is broad and found throughout the Hh gradient, it is tempting to speculate that the presence of this medial 'T' and subsequent low-affinity GBM could be an evolutionarily conserved mechanism used to generate variable levels of Hh target gene expression independent of a Hh threshold and distinct activator and repressor Gli isoforms.

## Interactions with other TFs control context-specific functions of Gli TFs in the face

While traditional descriptions of Gli-mediated Hh signal transduction do not include the requirement of binding partners, there is an established precedence for this concept. A number of TFs have been implicated as partners capable of interacting with Gli TFs and subsequently modulating Gli transcriptional activity. For example, Gli and Zic proteins were previously reported to physically interact through their zinc-finger domains to regulate subcellular localization and transcriptional activity important during neural and skeletal development (*Brewster et al., 1998*; *Koyabu et al., 2001*; *Zhu et al., 2008*). The pluripotency factor Nanog was also reported to physically interact with enhancer-bound Gli proteins to reduce the transcriptional response of cells to a Hh stimulus (*Li et al., 2016*). The Sox family of TFs has also been implicated in associating with Gli proteins to modulate transcriptional responses in various tissues (*Peterson et al., 2012*; *Tan et al., 2018*). Within the developing NT, Sox2 was determined to have a significant number of overlapping target genes, as Gli1 and Gli1/Sox2-bound CRMs were shown to induce Shh target gene expression (*Peterson et al., 2012*). Furthermore, Sox9 and Gli directly and cooperatively regulate several genes important in chondrocyte proliferation (*Tan et al., 2018*). Finally, recent studies have revealed that the bHLH TF Atoh1 synergizes with Gli2 to activate a medulloblastoma transcriptional network (*Yin et al., 2019*).

While several previous studies have reported interactions between Hand2 and the Gli TFs in the limb and in establishing left-right asymmetry, the mechanistic relationship appears to be tissue-specific and facets of this relationship still remain elusive. For example, Hand2 is believed to function downstream of Shh during establishment of left-right asymmetry (*Olson and Srivastava, 1996*), while Hand2 is believed to regulate *Shh* expression in the developing limb (*Charité et al., 2000*; *Fernandez-Teran et al., 2000*; *Yelon et al., 2000*; *McFadden et al., 2002*). Interestingly, in the context

of Hand2 acting upstream of Shh, previous studies suggested this to be a DNA-binding-independent effect, (*McFadden et al., 2002*) and propose that protein-protein interactions or dimer equilibrium can target Hand TFs to regulatory regions (*Firulli, 2003*).

Our work identified a novel relationship between Gli3 and Hand2 that is both unique to the tissue of origin (mandible) and the nature of the interaction (physical interaction, DNA-dependence) (*Figure 10*). First, our RNA-seq analyses on *Gli2^{f/f};Gli3^{f/f}*;Wnt1-Cre and *Hand2^{f/f}*-Wnt1-Cre mandibular tissue did not reveal any changes in *Hand2* or *Shh* expression, respectively, suggesting that unlike the relationship in the limb or in establishing polarity, there was not a cross-regulatory relationship between Shh and Hand2 in the mandible. Second, our site-directed mutagenesis experiments suggested that DNA-binding at some level is required for the Gli3/Hand2 synergism in the MNP, as opposed to the posited DNA-independent mechanism in the limb. Interestingly, the orientation and spacing of E-box and GBMs was not conserved, suggesting flexibility in the architecture underlying Gli3 and Hand2 co-regulatory interactions. The presence of additional TF motifs found in close proximity to GBMs, together with the established knowledge that Gli can interact with a number of other TFs, suggests that a larger protein complex may be at work (*Figure 10*, dotted circle). Furthermore, the cadre of proteins in this complex could vary depending upon the particular genomic locus and the role it plays regulating transcription either positively or negatively. Future studies will address the role, if any, these other proteins play in modulating Gli transcriptional output in the developing craniofacial complex.

## Gli3 functions as an activator within the developing craniofacial complex

In general, there are two accepted mechanisms for positive Gli-mediated transcriptional regulation: activation and de-repression (*Falkenstein and Vokes, 2014*). Activation refers to the full-length GliA isoform binding regulatory regions of target genes and driving gene expression. Gli1 and Gli2 play the predominant role in activating transcription (*Ding et al., 1998*; *Matise et al., 1998*; *Park et al., 2000*; *Stamataki et al., 2005*), and are believed to function within the highest concentrations of the Hh gradient (*Pan et al., 2006*). In the human face, loss of Gli2 has been associated with several craniofacial anomalies presenting with loss-of-function Hh phenotypes such as microcephaly, hypotelorism and a single central incisor (*Roessler et al., 2003*). Interestingly, mutations in Gli2 have no reported effects on the mandible.

De-repression is the second major mechanism of Hh signal transduction. In this case, targets silenced by the GliR require alleviation of this repression for expression. Subsequent activation can then occur from either GliA or additional TFs. GliR function is primarily carried out by Gli3 and is indispensable for proper Hh-dependent patterning (*Litingtung and Chiang, 2000*; *Wang et al., 2000*; *Oosterveen et al., 2012*; *Lex et al., 2020*). Recent studies in the limb, which is dependent upon Gli3R for proper patterning, have shown that not all GBMs are created equal. While some GBMs appear to be completely dependent on a Hh input, others remain 'stable' with Gli3 occupancy occurring independent of the morphogen (*Lex et al., 2020*). However, to date, no classification system of GBM utilization in the face has been established. In the face, Gli3R is also the predominant repressor for facial patterning, as loss of Gli3 has been associated with several human craniofacial anomalies presenting with gain-of-function Hh phenotypes including Greig cephalopolysyndactyly (*Vortkamp et al., 1991*; *Vortkamp et al., 1992*; *Hui and Joyner, 1993*; *Wild et al., 1997*).

Our current study reveals a previously unappreciated role for Gli3A in craniofacial development. Our genetic, biochemical and genomic data suggest Gli3/Hand2 complexes are specifically required to initiate patterning of the MNP and skeletogenic/glossogenic transcriptional networks. Several possibilities exist to explain why Hand2-dependent synergistic activation of targets may be unique to Gli3. First, while Gli3R is highly stable, Gli3A is reportedly not as stable as Gli2A (*Pan et al., 2006*; *Humke et al., 2010*; *Wen et al., 2010*). Association with Hand2 (and possibly other TFs in complex) may stabilize Gli3A, preventing degradation and allowing the isoform to function more efficiently. A second possibility is that Gli2A may predominantly utilize high-affinity, canonical GBMs to activate pathway targets, while Gli3A (when in complex with Hand2) predominantly utilizes low-affinity, divergent GBMs to activate tissue-specific targets independent of Hh concentration (*Figure 10*).

An additional explanation for the observed expression of *Gli3* outside the Shh threshold is that *Gli3* itself is additionally regulated by other molecules and functions independently of the Hh

pathway. A number of previous studies have shown that *Gli3* expression can be directly regulated by other pathways including the FGF and Wnt pathways (*Hasenpusch-Theil et al., 2012*; *Hasenpusch-Theil et al., 2017*). Given our findings, additional regulatory mechanisms may be at play in patterning the mandible that include both Hh-independent and Hh-dependent roles for Gli3, as have been described in the limb and thymus (*te Welscher et al., 2002*; *Hager-Theodorides et al., 2009*). Determining if these mechanisms also exist within the mandible, and within other craniofacial prominences, is one aspect of our ongoing work.

In closing, our results reveal a novel transcriptional mechanism for Gli signal transduction in the developing craniofacial complex outside of the traditional graded Hh signaling domains. Our data, compared to that in other organ systems, highlight the diversity of mechanisms utilized by Gli TFs across different tissues (*Figure 10*). As an organ system, the craniofacial complex is unique because it originates from facial prominences that constitute distinct developmental fields, in both cell content and transcriptional profiles. Thus, as Hand2 is only expressed in the MNP, our data pose the interesting possibility that facial prominences use unique, prominence-specific Gli partners to transduce Gli signals during craniofacial development. Furthermore, our data suggests sequence variation within GBMs, may also contribute to tissue-specific Gli transcriptional output. The discovery that a single base-pair within GBMs can relay significant transcriptional activity may lend new insight into examining genetic mutations in human patients with craniofacial anomalies.

# Materials and methods

## Key resources table

| Reagent type (species) or resource | Designation | Source or reference | Identifiers | Additional information |
|---|---|---|---|---|
| Antibody | Anti-Gli3 (Goat polyclonal) | R and D Systems | Cat.#AF3690; RRID:AB_2232499 | (1:1000) |
| Antibody | Anti-dHand M-19 (Goat polyclonal) | Santa Cruz Biotechnology | Cat.#Sc-9409; RRID:AB_2115995 | (1:1000) |
| Antibody | Anti-Flag M2 (Mouse monoclonal) | Sigma-Aldrich | Cat.#F1804; RRID:AB_262044 | (1:1000) |
| Cell line (*M. musculus*) | O9-1 cells | *Ishii et al., 2012* | | Gift from R. Lipinski Lab |
| Cell line (*E. coli*) | BL21 cells | Promega | Cat.#L1195 | |
| Peptide, recombinant protein | E47L/Tcf3 | Purified protein, contact B. Gebelein | NP_001157619.1 | peptide, recombinant protein |
| Peptide, recombinant protein | Gli3DBD | This paper | NP_032156.2 (full protein) | Purified protein, inquiries should be addressed to B. Gebelein |
| Peptide, recombinant protein | Hand2 | This paper | NP_034532.3 | Purified protein, inquiries should be addressed to B. Gebelein |
| Peptide, recombinant protein | Human FGF basic | R and D Systems | Cat.# 233-FB-025 | |
| Peptide, recombinant protein | Leukemia Inhibitory Factor (LIF) | Millipore | Cat.# ESG1106 | |
| Commercial assay or kit | Dual Luciferase Reporter Assay System | Promega | Cat.#E1910 | |
| Genetic reagent (*M. musculus*) | *Gli2^flox* | PMID:16571625 | JAX stock # 007926; RRID:IMSR_JAX:007926 | Gift from A. Joyner, Memorial-Sloan-Kettering Cancer Center |
| Genetic reagent (*M. musculus*) | *Gli3^flox* | PMID:18480159 | JAX stock # 008873; RRID:IMSR_JAX:008873 | |
| Genetic reagent (*M. musculus*) | *Hand2^flox* | PMID:17075884 | JAX stock # 027727; RRID:IMSR_JAX:027727 | |

*Continued on next page*

*Continued*

| Reagent type (species) or resource | Designation | Source or reference | Identifiers | Additional information |
|---|---|---|---|---|
| Genetic reagent (*M. musculus*) | *Wnt1-Cre: H2az2^{Tg(Wnt1-cre)11Rth}* | PMID:9843687 | MGI:2386570; RRID:IMSR_JAX:003829 | Gift from R. Stottmann |
| Genetic reagent (*M. musculus*) | *Gli3tm1.1Amc/ Grsr (3XFLAGbio)* | PMID:24990743; 27146892 | JAX stock #026135 | Generated by K.A. Peterson |
| Genetic reagent (*M. musculus*) | *Hand2^{3xFlag}* | PMID:25453830 | | Generated by R. Zeller |
| Recombinant DNA reagent | pGL3-promoter (plasmid) | Promega | Cat.#E1761 | Luciferase reporter |
| Recombinant DNA reagent | p3XFLAG-Gli3 (plasmid) | This paper | | Full-length Mouse Gli3 cloned into p3XFlag backbone. Inquiries should be addressed to S. Brugmann |
| Recombinant DNA reagent | p3XFLAG-Hand2 (plasmid) | This paper | | Full-length Mouse Hand2 cloned into p3XFlag backbone. Inquiries should be addressed to S. Brugmann |
| Sequence-based reagent | Gli2_Flox_F | This paper | PCR primers | AGG TCC TCT TAT TGT CAG GC; Inquiries should be addressed to S. Brugmann |
| Sequence-based reagent | Gli2_Flox_R | This paper | PCR primers | GAG ACT CCA AGG TAC TTA GC; Inquiries should be addressed to S. Brugmann |
| Sequence-based reagent | Gli3_Flox_F | This paper | PCR primers | GTC TGT AAC CAG ACG GCA CT; Inquiries should be addressed to S. Brugmann |
| Sequence-based reagent | Gli3_Flox_R | This paper | PCR primers | GAG AAT GTG TGA CTC CAT GC; Inquiries should be addressed to S. Brugmann |
| Sequence-based reagent | Hand2_Flox_F | JAX | PCR primers | ACT TGC TGA CTG GGT CCT TG; |
| Sequence-based reagent | Hand2_Flox_R | JAX | PCR primers | CTC GGC CTA GAG GAC ACT GA |
| Sequence-based reagent | Cre_F | This paper | PCR primers | GTCCCATTTA CTGACCGTAC ACC; Inquiries should be addressed to S. Brugmann |
| Sequence-based reagent | Cre_R | This paper | PCR primers | GTTATTCGGA TCATCAGCTA CACC; Inquiries should be addressed to S. Brugmann |
| Sequence-based reagent | 5'IREdye-700 labeled oligo | IDT | For EMSA assays | N/A |
| Sequence-based reagent | RNAscope probe- Mm Gli3 | Advanced Cell Diagnostics | Cat.#445511 | |
| Sequence-based reagent | RNAscope probe- Mm Hand2 | Advanced Cell Diagnostics | Cat.#499821 | |
| Sequence-based reagent | RNAscope probe- Mm Shh | Advanced Cell Diagnostics | Cat.#314361 | |
| Sequence-based reagent | RNAscope probe- Mm Ptch1 | Advanced Cell Diagnostics | Cat.#402811 | |
| Sequence-based reagent | RNAscope probe- Mm Foxd1 | Advanced Cell Diagnostics | Cat.#495501-C3 | |
| Sequence-based reagent | RNAscope probe- Mm Plagl1 | Advanced Cell Diagnostics | Cat.#462941 | |

*Continued on next page*

*Continued*

| Reagent type (species) or resource | Designation | Source or reference | Identifiers | Additional information |
|---|---|---|---|---|
| Sequence-based reagent | RNAscope probe- Mm Myh6 | Advanced Cell Diagnostics | Cat.#506251 | |
| Sequence-based reagent | RNAscope probe- Mm Gli2 | Advanced Cell Diagnostics | Cat.#405771 | |
| Sequence-based reagent | RNAscope probe- Mm Maf | Advanced Cell Diagnostics | Cat.#412951 | |
| Software, algorithm | Strand NGS | https://www.strand-ngs.com/ | | |
| Software, algorithm | HOMER | PMID:20513432 | RRID:SCR_010881 | |
| Software, algorithm | RELI | PMID:29662164 | | |
| Software, algorithm | COSMO | PMID:25905672 | | |
| Other | Cis-BP | PMID:25215497 | RRID:SCR_017236 | Transcription factor motif library http://cisbp.ccbr.utoronto.ca/index.php |

## Mouse strains

The *Wnt1-Cre*, *Hand2*$^{fl}$ (Stock No 027727), and *Gli3*$^{fl}$ (Stock No 008873) mouse strains were purchased from Jackson Laboratory. *Gli2*$^{f/f}$ mice were provided by Dr. Alexandra Joyner at Memorial Sloan-Kettering Cancer Center. As described in PMID 18501887, conditional deletion of *Hand2* using Wnt1-Cre is embryonic lethal ~E12 due to loss of norepinephrine. To rescue this phenotype and for investigation of *Hand2*$^{f/f}$;*Wnt1-Cre* mutants at later embryonic stages, beginning at embryonic day (E) 8, pregnant dams were fed water containing 100 µg/mL L-phenylephrine, 100 µg/mL isoproterenol, and 2 mg/mL ascorbic acid. All mice were maintained on a CD1 background. Both male and female mice were used. A maximum of 4 adult mice were housed per cage, and breeding cages housed one male paired with up to two females. All mouse usage was approved by the Institutional Animal Care and Use Committee (IACUC) and maintained by the Veterinary Services at Cincinnati Children's Hospital Medical Center. N $\geq$ 5 biologic replicates (biologically distinct samples) for each genotype shown.

## Embryo collection and tissue preparation

Timed matings were performed, with noon of the day a vaginal plug was discovered designated as E0.5. Embryos were harvested between E10.5–18.5, collected in PBS, and fixed in 4% paraformaldehyde (PFA) overnight at 4°C, unless otherwise noted. For paraffin embedding, embryos were dehydrated through an ethanol series, washed in xylene, and embedded in paraffin.

## Skeletal preparations

For skeletal preparations, E18.5 embryos were immersed in hot water before skin and soft tissue were removed. Embryos were immersed in 100% ethanol for 48 hr, then acetone for 48 hr. 0.015% alcian blue solution (20% glacial acetic acid and 80% 200 proof ethanol) for 24 hr to stain cartilage was added, then washed with ethanol for 24 hr. Embryos were immersed in 1% fresh KOH for 24–31 hr, then stained with 0.005% alizarin red (in 1% KOH) for 15 hr and transitioned through a series of glycerol dilutions.

## RNAscope *in situ* hybridization

Paraffin-embedded embryos were cut at 5 µm, and staining was performed with the RNAscope Multiplex Fluorescent Kit v2.0 according to the manufacturer's instructions. Briefly, sections were deparaffinized in xylene, rehydrated through an ethanol series, and antigen retrieval was performed. The following day, probes were hybridized to sections, paired with a fluorophore, and mounted with

Prolong Gold after counterstaining with DAPI. *Shh, Ptch1, Gli2, Gli3, Hand2, Foxd1, Myh6, Maf*, and *Plagl1* probes for the assay were designed and synthesized by Advanced Cell Diagnostics. RNA-Scope experiments were performed on N $\geq$ 3 biological replicates for each probe.

## RNA extraction and reverse transcription

RNA was extracted from cells using Trizol-Micro Total RNA Isolation Kit (Invitrogen, 15596026). cDNA was synthesized from up to 2 μg of RNA with the High Capacity RNA-to-cDNA Kit (Invitrogen, 4387406).

Quantitative Real-Time PCR qRT-PCR was performed in technical (multiple replicates of the same biological sample) triplicate using PowerUP SYBR Green Master Mix (ThermoFisher Scientific, A25742) on Applied Biosystems QuantStudio 3 Real-Time PCR System (ThermoFisher Scientific) for N = 3 biological replicates. All genes were normalized to *Gapdh* expression.

## Co-immunoprecipitation

MNPs were harvested from E10.5 CD-1 embryos, pooled, and lysed in RIPA buffer containing Halt protease inhibitor cocktail. Protein lysate was incubated with Hand2 (polyclonal goat IgG) or control goat IgG primary antibody overnight at 4C with nutation. Dynabeads Protein G were added the next day and incubated with antibody-lysate mixture for 4 hr at 4C on a nutator. Dynabeads Protein G-antibody-antigen complex was washed three times using RIPA buffer, and antigens were eluted from the beads in SDS sample buffer by boiling for 5 min. N = 4 biological replicates of pooled litters.

## Western blotting

For co-immunoprecipitation, eluted products and 10% of the input were separated by SDS-PAGE and transferred to a PVDF membrane for blotting at 4C with Gli3 (polyclonal goat IgG 1:1000, R and D Systems) and Hand2 (polyclonal goat IgG or mouse monoclonal IgG$_1$ 1:1000) primary antibodies. Detection of primary antibodies was performed using infrared-conjugated secondary antibodies (donkey anti-goat or goat anti-mouse IRDye 800CW, LICOR) and acquired using a LICOR infrared scanner. For plasmid verification, F primary antibody (monoclonal M2 mouse IgG$_1$) and enhanced chemiluminescence assay (Amersham ECL Primer, GE Healthcare Life Science) were used for detection.

## RNA-sequencing

MNPs were dissected from E10.5 embryos, using at three biologic samples. RNA was prepared for RNA-seq using Invitrogen RNAqueous-Micro RNA Isolation Kit (AM1931). Sequencing was carried out in 150 bp paired-end reads using the Illumina HiSeq2500 system.

## Single-cell RNA-sequencing

Mandibles from E11.5 or E13.5 wildtype CD1 mouse embryos were quickly dissected in ice-cold PBS and minced to a fine paste. Cells were dissociated into a single-cell suspension and sequenced using NovaSeq 6000 and the S2 flow cell. 12.5 mg of tissue was placed in a sterile 1.5 mL tube containing 0.5 mL protease solution containing 125 U/mL DNase and *Bacillus Licheniformis* (3 mg/mL for E11.5 sample and 5 mg/mL for E13.5 sample). The samples were incubated at 4C for a total of 10 min, with trituration using a wide boar pipette tip every minute after the first two. Protease was inactivated using ice-cold PBS containing 0.02% BSA and filtered using 30 μM filter. The cells were pelleted by centrifugation at 200G for 4 min and resuspended in 0.02% BSA in PBS. Cell number and viability were assessed using a hemocytometer and trypan blue staining. 9,600 cells were loaded onto a well on a 10x Chromium Single-Cell instrument (10X Genomics) to target sequencing of 6,000 cells. Barcoding, cDNA amplification, and library construction were performed using the Chromium Single-Cell 3' Library and Gel Bead Kit v3. Post cDNA amplification and cleanup was performed using SPRI select reagent (Beckman Coulter, Cat# B23318). Post cDNA amplification and post library construction quality control was performed using the Agilent Bioanalyzer High Sensitivity kit (Agilent 5067–4626). Libraries were sequenced using a NovaSeq 6000 and the S2 flow cell. Sequencing parameters used were: Read 1, 28 cycles; Index i7, eight cycles; Read 2, 91 cycles, producing about

300 million reads. The sequencing output data was processing using CellRanger (http://10xgenomics.com) to obtain a gene-cell data matrix.

## Chromatin immunoprecipitation

Individual ChIP-seq experiments were carried out on pooled embryonic tissue collected in ice-cold PBS. Dissected tissues were immediately fixed in 1% formaldehyde/PBS for 20 min at room temp followed by glycine quench (125 mM). ChIP procedures were performed as previously described (Peterson et al., 2012 and Osterwalder et al., 2014). All ChIP experiments were performed using mouse monoclonal anti-FLAG M2 antibody (Sigma-Aldrich). A mock control ChIP sample was made by performing ChIP on tissues isolated from wild-type embryos.

## ATAC-seq

Individual E11.5 MNP's were collected from wild-type embryos and immediately snap frozen in liquid nitrogen. Nuclei were isolated by incubating in homogenization buffer (250 mM sucrose; 25 mM KCl; 5 mM $MgCl_2$; 20 mM Tricine-KOH; 1 mM EDTA; and 1% IGEPAL) for 30 min at 4°C with shaking (800 rpm). Cell nuclei were counterstained with Trypan Blue and counted. Approximately $5 \times 10^4$ nuclei were processed for ATAC-seq as previously described (Buenrostro et al., 2015). DNA libraries were sequenced on NextSeq550 (Illumina) to generate 75 bp paired-end reads.

## Protein purification and EMSA

Coding regions for all protein fragments used for EMSA were cloned in-frame with an N-terminal 6xHis-tag in the pET14b vector (Novagen) and expressed in BL21 cells. The mouse E47 (E47L) isoform of the Tcf3 protein containing the bHLH domain (amino acids 271 to 648), the mouse Gli3 (Gli3DBD) protein containing the five zinc fingers in its DNA-binding domain (amino acids 465–648), and the full-length mouse Hand2 (Hand2FL) protein (amino acids 1–217) were purified under denaturing conditions via Ni-chromatography and refolded in Native lysis buffer while on Ni-beads as described previously (Witt et al., 2010; Zhang et al., 2019). Expression of each protein was confirmed via coomassie staining, and protein concentrations were measured via Bradford Assay. Probes were generated as previously described by annealing a 5'IREdye-700 labeled oligo from IDT with the following sequence 5'- CTATCGTAGACTTCG-3' to each oligo listed below and filling in via a Klenow reaction (Uhl et al., 2016). EMSAs were performed as previously described with the following modification to allow homodimer and heterodimer exchange between bHLH proteins (E47 and Hand2): binding reactions were incubated at 37°C for 40 min before allowing each reaction to cool to room temperature and incubating with DNA probes for an additional 15 min prior to separation on a native SDS gel (Uhl et al., 2010; Uhl et al., 2016). All EMSAs were imaged using a LICOR Clx scanner.

## *In vitro* cell culture

Immortalized O9-1 cranial NCCs were a gift from Dr. Robert Lipinski, originally provided by Dr. Robert Maxon, Keck School of Medicine at the University of Southern California. They were cultured as described in Ishii et al., 2012. Our lab confirmed the identity of these cells by qPCR of neural crest markers and by differentiation into neural crest derivatives. Cells were periodically screened to ensure no mycoplasma contamination.

## Plasmid constructs

Luciferase reporter constructs were generated by cloning putative enhancer fragments into the pGL3-promoter luciferase reporter plasmid. Hand2 and Gli3, were all cloned into a p3XFlag CMV 7.1 plasmid.

## Luciferase reporter assay

O9-1 cells were co-transfected in triplicate with the appropriate luciferase reporter plasmid, a Renilla control plasmid, and a combination of plasmids expressing Gli3 or Hand2 using Lipofectamine 3000. Cells were harvested 24 hr after transfection, and luciferase activity was determined using the Dual Luciferase Reporter Assay System (Promega) and the GLOMAX luminometer. N $\geq$ 3 biological replicates performed in technical triplicate for each condition.

## Bulk RNA-seq analysis

Paired-end reads were mapped to mm10 genome and transcript abundance was determined using Strand NGS. Differential expression was determined using DESeq2 within Strand NGS. Differentially expressed genes associated with GO-terms are listed in *Supplementary file 6*.

## Single-cell RNA-seq analysis

Raw reads were sequenced using 10x v2 chemistry for two samples E11.5 and E13.5 MNP. Reads were mapped to mouse transcriptome (mm10) version of the UCSC using Cellranger (*Zheng et al., 2017*, https://github.com/10XGenomics/cellranger). 7099 E11.5 cells and 6318 E13.5 cells were sequenced, with ~2300 genes per cell in the E11.5 sample, and ~2800 genes per cell in the E13.5 sample. Approximately 70% of the reads were confidently mapped to the transcriptome for each sample. Quality control (QC) was carried out where cells with less than ~1 k UMIs were removed from the quantification analysis. Finally, raw reads were quantified into a raw-counts matrix for cells that passed QC.

Raw counts matrix was analyzed using Seurat (v2.3.4) (*Stuart et al., 2019*). Briefly, all genes expressed in ≥3 cells and cells with at least 200 genes expressed were used for downstream analysis. Quality filtering of cells was done based on number of genes expressed and percent of mitochondrial expression. Followed by filtering, normalization of data was carried out using log2 transform and a global scaling factor. Highly variable genes (HVGs) which exhibit cell-to-cell variation, were selected by marking the outliers on average Expression vs dispersion plot and cell-cycle effect was regressed by removing the difference between the G2M and S phase. Next, HVGs were used to perform a linear dimension reduction using principal component analysis (PCA) and top 20 principal components (PCs) were used to cluster cells into respective clusters using graph-based knn clustering approach. Markers for each cluster were obtained using Wilcoxon rank sum test in 'FindAllMarkers' function. Cell clusters were annotated to respective cell types using a-priori knowledge of defined cell-type markers. Finally, clusters were visualized using t-distributed stochastic neighbor embedding (tSNE) a non-linear dimension reduction.

Further, to understand the similarities and differences among cell types annotated in each sample (E11.5, E13.5 MNP), an integration analysis was performed using Seurat (v3.0) (*Stuart et al., 2019*, https://github.com/Brugmann-Lab/Single-Cell-RNA-Seq-Analysis). Quality filtering, normalization, cell-cycle regression was performed as explained above. Feature selection (selecting HVGs) was done using variance stabilizing transform (vst) method as described in Seurat tutorial. Next, dimensionality reduction for both samples together was performed using diagonalized canonical correlation analysis (CCA) followed by L2-normalization and finally searching for mutual nearest neighbors (MNNs). Resulting cell-pairs from MNN were annotated as anchors ('FindIntegrationAnchors' function Seurat). Those integration anchors were then used to integrate the samples using 'IntegrateData' function in Seurat. After integrating the datasets, PCA was performed on integrated data, top 20 PCs were used for cell clustering using graph-based KNN algorithm and the clusters were visualized uniform manifold approximation projection (UMAP). All the visualization of the single-cell data was performed using data visualization functions embedded in Seurat.

Trajectory analysis or 'Pseudotime' analysis was performed using Monocle (*Trapnell et al., 2014*). Briefly, the integrated E11.5 and E13.5 scRNA-seq dataset was assessed for differential gene expression by original cluster, with the top 2000 being used for ordering. Data dimension reduction was performed using the DDRTree method, and cells were ordered using the orderCells function in Monocle 1. All visualization of the trajectory analysis was performed using functions embedded in Monocle.

## ChIP-seq analysis

ChIP-seq libraries were prepared according to manufacturers' instructions and $1 \times 75$ bp reads were generated on a NextSeq instrument (Illumina). The resulting reads were mapped to mouse genome assembly mm10 (GRCm38/mm10) using bwa (*Li and Durbin, 2009*). Pooled replicates were used to identify potential regulatory regions (*Supplementary file 4*). A final set of peak calls for each factor to use for motif enrichment was determined using bedtools (*Quinlan and Hall, 2010*) to merge biological replicates and identify peaks shared between replicates (*Supplementary file 5*). ChIP-seq peak overlap significance was calculated using the RELI software package (*Harley et al.,*

*2018*; https://github.com/WeirauchLab/RELI). Nearest upstream and downstream genes were determined for each ChIP-seq peak for global analysis and comparison to bulk and scRNA-seq datasets. Gli3/Hand2 overlapping ChIP-seq peaks were also split into the following categories: those with a **c**GBM (80% match to the top CisBP identified canonical GBM, M08023_2.00), those without a **c**GMB, and those without a **c**GMB and with a **d**GBM (with at least CCTCC). TF binding site motif enrichment analyses were performed using the HOMER software package (*Heinz et al., 2010*) modified to use a log 2-based scoring system and contain mouse motifs obtained from the Cis-BP database, build 1.94d (*Weirauch et al., 2014*). DNA 8mer counts were calculated by examining the number of times each of the possible 32,896 8mers occurs in the sequences contained within the given ChIP-seq peakset (on either strand, avoiding double-counting for palindromic sequences). Enrichment for particular orientations and spacings between Gli and Hand motifs was performed using the COSMO software package (*Narasimhan et al., 2015*).

Statistical Analysis qPCR and luciferase data are represented as mean + SD. Relative luciferase output was calculated by normalizing raw Luciferase output to Renilla output and comparing this dual luciferase output to a control condition. Statistical significance was determined using Student's *t* test. p-value<0.05 was considered statistically significant. *p<0.05, **p<0.01, and ***p<0.001.

## Acknowledgements

The authors would like to acknowledge the CCHMC DNA Sequencing Core, Gene Expression Core, and Veterinary Services Cores, the Genome Technologies services at The Jackson Laboratory, and the Center for Epigenomics at the University of California San Diego. AZ and RZ thank Jens Stolte for expert technical assistance. We acknowledge support from the National Institute of Health, National Institute for Dental and Craniofacial Research to SAB (R35 DE027557) and KHE (F31 DE027872), National Institute of General Medical Sciences to KAP (R01 GM124251) and BG (R01 GM079428), and Cincinnati Children's Hospital CpG and Endowed Scholar awards to MTW.

## Additional information

### Funding

| Funder | Grant reference number | Author |
|---|---|---|
| National Institutes of Health | R35DE027557 | Samantha A Brugmann |
| National Institutes of Health | R01GM124251 | Kevin A Peterson |
| National Institutes of Health | F31DE027872 | Kelsey H Elliott |
| National Institutes of Health | R01GM079428 | Brian Gebelein |

The funders had no role in study design, data collection and interpretation, or the decision to submit the work for publication.

### Author contributions

Kelsey H Elliott, Data curation, Formal analysis, Supervision, Funding acquisition, Validation, Investigation, Visualization, Writing - original draft, Writing - review and editing; Xiaoting Chen, Resources, Data curation, Software, Formal analysis; Joseph Salomone, Validation, Investigation; Praneet Chaturvedi, Data curation, Software, Investigation; Preston A Schultz, Sai K Balchand, Jeffrey D Servetas, Data curation, Investigation; Aimée Zuniga, Rolf Zeller, Resources, Investigation; Brian Gebelein, Data curation, Formal analysis, Supervision, Methodology, Writing - review and editing; Matthew T Weirauch, Resources, Data curation, Software, Formal analysis, Supervision, Investigation, Methodology, Writing - review and editing; Kevin A Peterson, Resources, Data curation, Software, Formal analysis, Supervision, Funding acquisition, Validation, Investigation, Methodology, Project administration, Writing - review and editing; Samantha A Brugmann, Conceptualization, Resources, Supervision, Funding acquisition, Writing - original draft, Project administration, Writing - review and editing

## Author ORCIDs
Brian Gebelein  http://orcid.org/0000-0001-9791-9061
Samantha A Brugmann  https://orcid.org/0000-0002-6860-6450

## Ethics
Animal experimentation: This study was performed in strict accordance with the recommendations in the Guide for the Care and Use of Laboratory Animals of the National Institutes of Health. All of the animals were handled according to approved institutional animal care and use committee (IACUC) protocols (IACUC2017-0063) of Cincinnati Children's Hospital Medical Center.

## Decision letter and Author response
Decision letter https://doi.org/10.7554/eLife.56450.sa1
Author response https://doi.org/10.7554/eLife.56450.sa2

# Additional files
## Supplementary files
• Supplementary file 1. Differences in gene expression levels from conditional KO bulk RNA-seq.

• Supplementary file 2. Results from Simple counting method of quantifying instances of GBMs in ChIP-seq data.

• Supplementary file 3. Results from COSMO algorithm. Number of instances of GBM and E-box spacing and orientation varieties in Gli3-Hand2 overlap peaks.

• Supplementary file 4. Pooled ChIP-seq replicate peak calls.

• Supplementary file 5. Shared ChIP-seq peaks between replicates.

• Supplementary file 6. GO-terms associated with Differentially Expressed Genes.

• Transparent reporting form

## Data availability
Sequencing data have been deposited in GEO under accession codes GSE141431, GSE141173. ChIP data have been deposited in GEO under accession code GSE146961 All data generated or analyzed during this study are included in the manuscript and supporting files. Source data files have been provided for Figures 1,5,8,9, Figure 1—figure supplement 1, Figure 8—figure supplement 2, and Figure 9—figure supplement 1.

The following datasets were generated:

| Author(s) | Year | Dataset title | Dataset URL | Database and Identifier |
| --- | --- | --- | --- | --- |
| Elliott KH, Brugmann SA | 2020 | bulk RNA-seq of wild-type, Gli2f/f; Gli3f/f;Wnt1-Cre, and Hand2f/f; Wnt1-Cre mouse e10.5 dissection mandibular prominences | https://www.ncbi.nlm.nih.gov/geo/query/acc.cgi?acc=GSE141431 | NCBI Gene Expression Omnibus, GSE141431 |
| Elliott KH, Brugmann SA | 2020 | Single cell sequencing of dissected mouse mandibular prominence at embryonic day e11.5 and e13.5 | https://www.ncbi.nlm.nih.gov/geo/query/acc.cgi?acc=GSE141173 | NCBI Gene Expression Omnibus, GSE141173 |
| Elliott KH, Brugmann SA, Peterson KA | 2020 | Gli3 and Hand2 ChIP-sequencing of developing face and mandibular prominence | https://www.ncbi.nlm.nih.gov/geo/query/acc.cgi?acc=GSE146961 | NCBI Gene Expression Omnibus, GSE146961 |

The following previously published dataset was used:

| Author(s) | Year | Dataset title | Dataset URL | Database and Identifier |
| --- | --- | --- | --- | --- |
| Osterwalder M, Speziale D, Shoukry M, Mohan R, Iva- | 2014 | Genome-wide candidate HAND2 target regions in mouse embryonic tissues | https://www.ncbi.nlm.nih.gov/geo/query/acc.cgi?acc=GSE55707 | NCBI Gene Expression Omnibus, GSE55707 |

nek R, Kohler M,
Beisel C, Wen X,
Scales SJ, Christof-
fels VM

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
