## [Decision Letter]

**Acceptance summary:**

This paper demonstrates co-operation between Gli factors and Hand2 in the development of the mandibular prominence and the initiation of transcriptional networks required for neural crest cell differentiation into distinct lineages. The work distinguishes the functional consequences of utilizing a canonical Gli binding motif vs. a divergent Gli binding motif, making a novel contribution to understanding craniofacial development.

**Decision letter after peer review:**

Thank you for submitting your article "Gli3 utilizes Hand2 to synergistically regulate tissue-specific transcriptional networks" for consideration by *eLife*. Your article has been reviewed by three peer reviewers, and the evaluation has been overseen by a Guest Editor and Kathryn Cheah as the Senior Editor. The reviewers have opted to remain anonymous.

The reviewers have discussed the reviews with one another and the Guest Editor has drafted this decision to help you prepare a revised submission.

Summary:

This study describes a novel mechanism of developmental regulation by synergy between Gli3/2 and Hand2 in craniofacial development. In an elegant extensive study employing a combination of single-cell transcriptomic characterisation of mandibular prominence (MNP) markers, mouse conditional knock-out models for Gli2/3 and Hand2, chromatin binding maps and identification and validation of putative regulatory elements, they show that Gli3 cooperates with Hand2 to drive a new set of targets in the developing MNPs that is independent of a HH morphogen gradient. The broader implication of the work is that synergistic cooperation between GLI and HAND factors may operate to regulate other developmental processes.

1) Better integration across datasets and reanalyses of the data with statistical correlation to strengthen the conclusions and provide in-depth insight into mechanistic consequences and possible non-canonical Gli3 activity.

2) Overstatements should be tempered and mis-statements corrected with respect to correlating phenotype to the proposed model.

Revisions expected in follow-up work:

1) Improved bioinformatics analyses, in particular combining single-cell co-expression, binding maps and the identified regulatory elements and integration of the presented datasets.

While no further experiments should be required, some aspects of the analysis that are insufficiently developed should be addressed in the revision.

No additional datasets need to be produced, but further analysis and statistical correlation and integration of multiple datasets are required.

2) Single-cell analysis of MNPs indicates heterogeneous expression of Gli factors and Hand2 in different subclusters. Thus, the presented feature plots do not appropriately reflect the gene co-expression data, and heatmaps, focussing on a limited number of factors (Gli2, Gli3, Hand2 and several canonical and non-canonical targets) should be presented.

3) As part of the analysis of 11.5 and 13.5 scRNA-seq MNP data, the statements about E13.5 muscle and osteogenic clusters deriving from E11.5 Gli3^+^/Hand2^+^ NCC clusters should be substantiated using future cell state prediction by trajectory analysis of integrated single-cell data.

4) The overall impression is that while this study contributes a significant amount of new data, those are greatly under-analysed and stronger, better support and conclusions could be drawn upon reanalysis. In particular, ChIP and associated ATAC datasets are underexploited, there is no link to downstream targets globally, and no definition in which sc-cluster the targeted genes are present/enriched. Furthermore, no global investigation and statistical analysis of a possible correlation between the Gli3/Hand2 binding and the co-expression with non-canonical targets is presented. Also, analysis of KO data should be integrated into this interpretation, and, again, statistical correlation between datasets established. Better integration across datasets will afford in-depth insight into mechanistic consequences and possible non-canonical Gli3 activity, resulting in greater overall impact of this work.

5) The authors don't need to overstate the phenoytpic and GO biological process similarity for the Hand2/Gli2/3;Wnt1-Cre compound het mice, to justify their subsequent model which is subsequently well justified. The authors should however be more precise with their anatomical comparisons in establishing the rationale for their studies.

---

## [Author Response]

Revisions for this paper:1) Better integration across datasets and reanalyses of the data with statistical correlation to strengthen the conclusions and provide in-depth insight into mechanistic consequences and possible non-canonical Gli3 activity.

We greatly appreciate the reviewers comments and agree that better integration across datasets was necessary. We have added a significant amount of new data analyses to address this concern. First, comparison of bulk RNA-seq and ChIP-seq data revealed that among the differentially expressed genes in the Gli2/3 conditional knock-outs, more had proximal Gli3/Hand2 overlapping peaks than Gli3 peaks without overlapping Hand2 peaks (Figure 4B; subsection “Gli3 and Hand2 occupy CRMs near shared targets in mandibular neural crest cells”). Furthermore, using our RELI (Regulatory Element Locus Intersection) tool, we tested the enrichment of Gli3 alone or Gli3/Hand2 overlapping peaks proximal to scRNA-seq cluster marker genes. This analysis revealed a significant enrichment of Gli3 alone peaks in cluster marker genes of the MNP, but not specifically in clusters 0, 4, 5 which correspond to NCC-derived skeletal and glossal musculature progenitors (Figure 4C). Interestingly, and supportive of our hypotheses, when we utilized RELI to test enrichment of scRNA-seq cluster markers closest to Gli3/Hand2 overlapping peaks, we found that there was significant enrichment in cluster marker genes of the MNP, and highest enrichment in clusters 0, 4, and 5 (Figure 4C; subsection “Gli3 and Hand2 occupy CRMs near shared targets in mandibular neural crest cells”). Thus, this analysis supports a biologically relevant role for Gli3 and Hand2 interactions to regulate target genes within the developing MNP.

Our initial analysis revealed that 50% of genes differentially expressed in Gli2/3 cKO MNPs were also differentially expressed in Hand2 cKO MNPs (Figure 3C). We further analyzed these data to show that within that 50%, 29% were decreased in both mutants and 21% were increased in both mutants (Figure 3—figure supplement 1A-A’; subsection “Gli3 and Hand2 occupy CRMs near shared targets in mandibular neural crest cells”). We repeated these analyses for genes that had Gli3/Hand2 overlapping peaks. 17% of genes with a Gli3/Hand2 overlapping peak that were increased in the Gli2/3 cKO, were also increased in the Hand2 cKO. Conversely, 33% of genes with a Gli3/Hand2 overlapping peak that were decreased in the Gli2/3 cKO, were also decreased in the Hand2 cKO (Figure 4—figure supplement 1A; subsection “Gli3 and Hand2 occupy CRMs near shared targets in mandibular neural crest cells”).

We next integrated our ChIP-seq and scRNA-seq data to determine if there was a functional consequence of Gli3-Hand2 interactions. To this end, we examined marker genes for neural crest cell (NCC) clusters, asking if they were associated with Gli3 peaks alone or Gli3/Hand2 overlapping peaks (Figure 4—figure supplement 1B; subsection “Gli3 and Hand2 occupy CRMs near shared targets in mandibular neural crest cells”). These analyses revealed that marker genes for NCCs had statistically stronger association with Gli3/Hand2 overlapping peaks compared to Gli3 alone peaks. Furthermore, genes that are differentially expressed in both Gli2/3 and Hand2 conditional KOs are more likely to be marker genes for clusters 0, 4 and 5 than marker genes for other NCC clusters or other non-NCC clusters (Figure 4—figure supplement 1C; subsection “Gli3 and Hand2 occupy CRMs near shared targets in mandibular neural crest cells”).

We further analyzed GO-terms associated with differentially expressed genes with either proximal Gli3 alone peaks or Gli3-Hand2 overlapping peaks to specifically address possible mechanistic consequences of a Gli3-Hand2 interaction, as suggested by the reviewers. Overall, the GO-terms for differentially expressed genes with Gli3 alone peaks were substantially different from the GO-terms for differentially expressed genes with Gli3/Hand2 overlapping peaks (Figure 4A; subsection “Gli3 and Hand2 occupy CRMs near shared targets in mandibular neural crest cells”). Interestingly, while GO-terms for differentially expressed genes with Gli3 alone peaks included pattern specification, embryonic organ development and Hh signaling, those associated with differentially expressed genes with overlapping Gli3/Hand2 peaks included a different set of processes including chondrocyte differentiation and muscle cell differentiation (Figure 4A; subsection “Gli3 and Hand2 occupy CRMs near shared targets in mandibular neural crest cells”). Together, these integrative analyses support a distinct mechanistic role for Gli3-Hand2 interactions vs. those influenced by Gli3 alone.

In summary in the new Figure 4 and the accompanying supplement (Figure 4—figure supplement 1) served to profile the functional differences between a Gli3 alone binding vs. Gli3/Hand2 cooperation. Specifically, we show:

1) Gli3 alone vs. Gli3/Hand2 binding events are associated with the differential expression of target genes implicated in different biological processes (Figure 4A).

2) There is more potential Gli3/Hand2 cooperation in NCC clusters (Figure 4—figure supplement 1B; ChIP-seq peaks compared to single cell).

3) Genes that require Gli3 and Hand2 for their expression are more likely to be markers of NCC clusters 0, 4, and 5 (Figure 4—figure supplement 1C; Bulk RNA-seq compared to single cell RNA-seq).

4) Clusters 0, 4, and 5 give rise to skeletogenic and glossal musculature derivatives (Figure 4E-F).

In our next set of computational experiments, we went on to further analyze the function of Gli3/Hand2 interactions and examined the possible functional consequences of Gli3 using a canonical DNA-binding site vs. the divergent binding site when interacting with Hand2.

We first analyzed GO-terms associated with differentially expressed genes with Gli3/Hand2 overlapping peaks to specifically address possible functional consequences of utilization of a canonical or divergent GBM (cGBM or dGBM; Figure 5—figure supplement 1A; subsection “ Low-affinity Gli binding motifs are within close proximity to E-boxes and specific to the developing mandible” and Figure 6A; subsection “dGBMs direct unique gene regulatory programs in neural-crest-derived skeletal and glossal progenitors of the MNP”). Overall, the GO-terms associated with cGBMs were substantially different from those associated with dGBMs. Furthermore, GO-terms associated specifically with the dGBM represented a muscle-specific subset.

We further compared our bulk RNA-seq data to our ChIP-seq data and found that of the differentially expressed (DE) genes in both Gli2/3 and Hand2 conditional KOs more DE genes were associated with dGBMs rather than cGBMs (Figure 6B; subsection “dGBMs direct unique gene regulatory programs in neural-crest-derived skeletal and glossal progenitors of the MNP”). These data suggested that dGBMs were prevalent within Gli3/Hand2 overlapping peaks associated with DE genes in both conditional mutants.

To address the functional consequences of utilizing a cGBM or dGBM, we compared our scRNA-seq and ChIP-seq data for Gli3-Hand2 overlapping peaks and used GO-terms to identify enriched biological processes. These analyses suggest that cGBMs are significantly associated with neurogenic biological processes, whereas dGBMs are specifically associated with skeletogenic biological processes (Figure 6C; subsection “dGBMs direct unique gene regulatory programs in neural-crest-derived skeletal and glossal progenitors of the MNP”).

We next addressed the possible biological significance of cGBM or dGBM. We used RELI to determine that there is significant enrichment of dGBMs within NCC clusters (Figure 6D; subsection “dGBMs direct unique gene regulatory programs in neural-crest-derived skeletal and glossal progenitors of the MNP”). Furthermore, we determined that dGBMs were enriched near genes marking NCC clusters, when compared to all other clusters (Figure 6E; subsection “dGBMs direct unique gene regulatory programs in neural-crest-derived skeletal and glossal progenitors of the MNP”).

2) Overstatements should be tempered and mis-statements corrected with respect to correlating phenotype to the proposed model.

We significantly edited our statements related to the characterization of skeletal phenotypes in Gli2/3 and Hand2 cKOs (subsection “Loss of Gli TFs and Hand2 generates micrognathia and aglossia”). We now conclude that these data suggest that while Gli2/3-mediated networks may function in proximal mandibular patterning and Hand2 mediated networks may function primarily in distal mandibular patterning, it is unclear if these networks interact or overlap with one another during mandibular development. We also tempered statements about the triple heterozygotes to emphasize that phenotypes were indeed subtle (“…triple heterozygotes (Hand2^f/+^;Gli2^f/+^;Gli3^f/+^;Wnt1-Cre) resulted in subtle yet significant MNP phenotypes, including low-set pinnae, micrognathia, smaller incisors, and hypoglossia.”). We have also moved the triple heterozygous data to Figure 1—figure supplement 1G-H’, I and added data for the triple homozygous mutants to Figure 1G--I and Figure 1—figure supplement 1D. We based our subsequent experiments on the hypothesis that Gli2/3 and Hand2 are working together during mandibular development rather than in parallel.

Revisions expected in follow-up work:1) Improved bioinformatics analyses, in particular combining single-cell co-expression, binding maps and the identified regulatory elements and integration of the presented datasets.

We again thank the reviewers for this suggestion, as the integration of our datasets strengthens our previous conclusions. As the comprehensive list of additional analyses is described under point #1 of Revisions for this paper (above), here we specifically outline the new improved bioinformatic analyses with our single cell RNA-seq (e.g., (1) single-cell (sc)RNA-seq and bulk RNA-seq data and (2) scRNA-seq and ChIP-seq data).

1) Comparing bulk RNA-seq from conditional mutants to single cell RNA-seq, we found that the highest enrichment and percentage of genes differentially expressed in conditional mutants were the marker genes from clusters 0, 4 and 5 (Figure 4—figure supplement 1C; subsection “Gli3 and Hand2 occupy CRMs near shared targets in mandibular neural crest cells”). While this was not significant against marker genes for other neural crest cell clusters, it was significant against marker genes for all other non-neural crest cell clusters.

2) By using RELI to test enrichment of scRNA-seq cluster marker genes closest to Gli3 alone or Gli3-Hand2 overlapping peaks, we found that there was not significant enrichment of Gli3 alone peaks associated with clusters 0, 4 and 5 (Figure 4C; subsection “Gli3 and Hand2 occupy CRMs near shared targets in mandibular neural crest cells”). In contrast, we found that there was significant enrichment of Gli3-Hand2 overlapping peaks associated with clusters 0, 4 and 5 (Figure 4C; subsection “Gli3 and Hand2 occupy CRMs near shared targets in mandibular neural crest cells”). These results were supported when comparing intersection of neural crest cell markers compared to non-neural crest cells (Figure 4—figure supplement 1B; subsection “Gli3 and Hand2 occupy CRMs near shared targets in mandibular neural crest cells”). Thus, this analysis supports a biologically relevant role for Gli3 and Hand2 interactions to regulate target genes within the developing MNP, and specifically in the developing skeletal and glossal derivatives (originating from E11.5 clusters 0, 4, and 5).

Furthermore, the RELI algorithm revealed a significant enrichment of dGBMs in marker genes associated with NCC clusters (Figure 6D; subsection “dGBMs direct unique gene regulatory programs in neural-crest-derived skeletal and glossal progenitors of the MNP”). Additionally, there was significant enrichment of Gli3-Hand2 overlapping peaks that contained a dGBM associated with NCC cluster marker genes vs. cluster markers for other cell types (Figure 6E – integration of scRNA-seq and motif enrichment from ChIP-seq; subsection “dGBMs direct unique gene regulatory programs in neural-crest-derived skeletal and glossal progenitors of the MNP”).

While no further experiments should be required, some aspects of the analysis that are insufficiently developed should be addressed in the revision.No additional datasets need to be produced, but further analysis and statistical correlation and integration of multiple datasets are required.2) Single-cell analysis of MNPs indicates heterogeneous expression of Gli factors and Hand2 in different subclusters. Thus, the presented feature plots do not appropriately reflect the gene co-expression data, and heatmaps, focussing on a limited number of factors (Gli2, Gli3, Hand2 and several canonical and non-canonical targets) should be presented.

We completely agree with reviewers that our original presentation of data did not properly portray our conclusion. We have edited the text and/or added data to both figures (old Figures 2 and 4) that presented feature plots. In the revised manuscript, we added additional analyses and rationale to explain why we focus on clusters 0, 4, and 5 (subsection “Gli2, Gli3, and Hand2 are co-expressed in NCC-derived populations associated with skeletal and glossal progenitors” and Figure 2—figure supplement 1D, E). We used RNAscope to illustrate the evenly distributed and widespread expression of Gli3 (Figure 2A-C’). This is confirmed with feature plots from E11.5 (Figure 2I). We added analyses revealing that 49% of cells with robust Gli3 expression (greater than 1.5 tpm) occupy clusters 0, 4, and 5. Furthermore, 50% of Gli3^+^ cells in clusters 0, 4 and 5 were also Hand2^+^ (subsection “Gli2, Gli3, and Hand2 are co-expressed in NCC-derived populations associated with skeletal and glossal progenitors” and Figure 2—figure supplement 1D, E). We further show more confined expression of Hand2 in the medial aspect of the MNP (Figure 2D-F’). This was also confirmed with feature plots from E11.5 MNP cells. We now add additional analyses revealing that 62% of cells with robust Hand2 expression (greater than 1.5 tpm) occupy clusters 0, 4, and 5 (Figure 2—figure supplement 1D, E) and that while clusters 0, 4 and 5 account for only 31% of cells in the MNP, they account for 43% of all Hand2 expressing cells. This finding was in stark contrast to other clusters with robust Hand2 expression (clusters 7, 8, 16), which only account for 14% of the highest expressing Hand2 cells (subsection “Gli2, Gli3, and Hand2 are co-expressed in NCC-derived populations associated with skeletal and glossal progenitors”). Thus, the overlapping expression (confirmed by both RNAscope and scRNA-seq) and contribution of genes expressed in/phenotypes associated with clusters 0, 4 and 5, lead us to focus our subsequent analyses on the relationship between Gli3 and Hand2 within these clusters. We do not suggest that Gli3 and Hand2 only overlap or interact in clusters 0,4, and 5. Instead, we simply focus on how they function in clusters 0,4, and 5 because those clusters give rise to tissues associated with the mandibular phenotypes observed in the conditional knockouts.

When comparing scRNA-seq data from E11.5 and 13.5 MNPs (old Figure 4), our use of feature plots was not to suggest that Gli factors, Hand2 and targets maintained expression in highlighted E13.5 clusters, but rather to show that identified target genes were expressed in clusters that originated from E11.5 clusters 0, 4 and 5. Thus, our data suggest that Gli factors and Hand2 cooperate at early stages of MNP development (E11.5) to initiate transcriptional networks necessary for the differentiation of NCCs into skeletal and glossal cell types present at E13.5. We used RNAscope on early mandibular sections to illustrate that these target genes, despite being considered Gli-targets (differentially expressed in conditional KO and ChIP-seq peaks) were expressed in areas outside the highest Ptch1^+^ expression domain. Thus, these data suggest that the widely accepted model of a Hh gradient was not sufficient to explain Gli function in the developing MNP. To clarify these points, we significantly added to and restructured our previous figures to make a new Figure 7 and Figure 7—figure supplement 1. The new Figure 7 (subsection “dGBMs direct unique gene regulatory programs in neural-crest-derived skeletal and glossal progenitors of the MNP”) shows that a number of target genes (all of which are differentially expressed in Gli2/3 conditional knockouts and have overlapping Gli3/Hand2 peaks with a dGBM) are expressed in E11.5 clusters 0, 4, and 5, and in E13.5 clusters that originate from E11.5 clusters 0, 4, and 5. This conclusion is supported by the new trajectory analysis in Figure 4E, F, which we highlight by E13.5 cluster in Figure 7M-S. In summary, cluster expression of those target genes at E11.5 (Figure 7E-H), cluster expression of target genes at E13.5 (Figure 7I-L), trajectory analysis of E11.5 clusters to E13.5 clusters (Figure 7M-S), and finally we show target gene expression in the early mandible (Figure 7V-Y).

We did try to generate heatmaps as suggested by the reviewers, but they did not clearly illustrate our point. Instead, we used our E11.5 scRNA-seq data to quantify the percentage of NCCs expressing either Gli3 or Hand2, and our chosen targets (Figure 7—figure supplement 1B; subsection “dGBMs direct unique gene regulatory programs in neural-crest-derived skeletal and glossal progenitors of the MNP”). Furthermore, we also examined the percentage of cells that did not express Ptch1 but did express these target genes (Figure 7—figure supplement 1B, Ptch1-; subsection “dGBMs direct unique gene regulatory programs in neural-crest-derived skeletal and glossal progenitors of the MNP”). Together, these new data support that:

1) Gli3 and Hand2 are co-expressed in early neural crest cells (E11.5 clusters 0, 4, and 5; Figure 2I-J).

2) Clusters 0, 4, and 5 ultimately differentiate into skeletal and glossal derivatives (E13.5 clusters 1, 3, 4, 11, 15, and 19; Figure 7I-S).

3) While neither RNA-scope at 10.5 (Figure 7T-Y) or scRNA-seq at 11.5 (Figure 7E-H) is completely capable of revealing the full extent of Gli3^+^, Hand2^+^, target gene + and Ptch1- cells over the course of MNP development (E9-13.5), we detected a percentage of cells that did meet all these criteria (Figure 7—figure supplement 1B), further supporting the argument that this novel transcriptional mechanism is at play during mandibular development.

3) As part of the analysis of 11.5 and 13.5 scRNA-seq MNP data, the statements about E13.5 muscle and osteogenic clusters deriving from E11.5 Gli3^+^/Hand2 NCC clusters should be substantiated using future cell state prediction by trajectory analysis of integrated single-cell data.

We have performed future cell state predictions by trajectory analysis of the integrated single-cell RNA-seq dataset using Monocle. These data are now presented in Figure 4E, F (subsection “Gli3 and Hand2 occupy CRMs near shared targets in mandibular neural crest cells”). Specifically, in Figure 4F these analyses reveal that E11.5 clusters 0, 4 and 5 give rise to E13.5 clusters 1, 15, 19 (glossal) and 3, 4, 5, 6, and 11 (skeletogenic) (Figure 7M-S, subsection “dGBMs direct unique gene regulatory programs in neural-crest-derived skeletal and glossal progenitors of the MNP”). We have moved our previous data (integrated UMAPs) to Figure 4—figure supplement 1. Thus, these new data substantiated previous data showing that Gli3 and Hand2 cooperation at E11.5 in clusters 0, 4 and 5 are necessary for initiating transcriptional networks necessary for skeletogenic and glossal development.

4) The overall impression is that while this study contributes a significant amount of new data, those are greatly under-analysed and stronger, better support and conclusions could be drawn upon reanalysis. In particular, ChIP and associated ATAC datasets are underexploited, there is no link to downstream targets globally, and no definition in which sc-cluster the targeted genes are present/enriched.

We again thank the reviewers for this comment and have performed several additional analyses to address global trends in our datasets. In particular, we globally examined ChIP-seq and bulk RNA-seq targets and assessed which gene classes had Gli3 alone vs. Gli3/Hand2 overlapping peaks. These data provided evidence for enrichment of Gli3/Hand2 overlapping peaks within a substantially distinct class of genes relative to Gli3 alone peaks (Figure 4A; subsection “Gli3 and Hand2 occupy CRMs near shared targets in mandibular neural crest cells”). We also globally examined scRNA-seq cluster markers near Gli3/Hand2 overlapping peaks and assessed which gene classes had cGBMs of dGBMs. These data provided evidence for increased prevalence of dGBMs associated with marker genes implicated in distinct biological processes (Figure 6C; subsection “dGBMs direct unique gene regulatory programs in neural-crest-derived skeletal and glossal progenitors of the MNP”).

We further examined scRNA-seq cluster enrichment using the RELI algorithm. This analysis revealed that while there was significant enrichment of cluster marker genes associated with the entire E11.5 MNP near Gli3 alone peaks, there was not a significant enrichment for marker genes for only clusters 0, 4, 5 (Figure 4C, left; subsection “Gli3 and Hand2 occupy CRMs near shared targets in mandibular neural crest cells”). Conversely, while overlapping Gli3/Hand2 peaks were also enriched in all MNP clusters, they had greater enrichment in clusters 0, 4, and 5 (Figure 4C, right; subsection “Gli3 and Hand2 occupy CRMs near shared targets in mandibular neural crest cells”). Additionally, we examined the percent of cluster markers according to GBM class. We found that dGBMs were significantly enriched near NCC cluster marker genes (Figure 6D- RELI analysis; subsection “dGBMs direct unique gene regulatory programs in neural-crest-derived skeletal and glossal progenitors of the MNP”). Furthermore, there were significantly more NCC cluster markers with dGBMs than with cGBMs (Figure 6E; subsection “dGBMs direct unique gene regulatory programs in neural-crest-derived skeletal and glossal progenitors of the MNP”).

The inclusion of ATAC-seq data in this manuscript was to confirm that of Gli3/Hand2 overlapping peaks were within areas of open chromatin, and as such were capable of transcription. Further analyses using the ATAC-seq data set is the focus of ongoing work to identify regulatory regions within the MNP. As such we feel this is outside the scope of this project.

Furthermore, no global investigation and statistical analysis of a possible correlation between the Gli3/Hand2 binding and the co-expression with non-canonical targets is presented. Also, analysis of KO data should be integrated into this interpretation, and, again, statistical correlation between datasets established. Better integration across datasets will afford in-depth insight into mechanistic consequences and possible non-canonical Gli3 activity, resulting in greater overall impact of this work.We thank reviewers for suggesting increased global investigation of the outcome of Gli3/Hand2 binding. We have addressed these critiques above in point 1 under the Revisions for this paper section.5) The authors don't need to overstate the phenoytpic and GO biological process similarity for the Hand2/Gli2/3;Wnt1-Cre compound het mice, to justify their subsequent model which is subsequently well justified. The authors should however be more precise with their anatomical comparisons in establishing the rationale for their studies.

We agree with the reviewers that the compound heterozygous mutants were not the best justification, as such we have moved these data to Figure 1—figure supplement 1. We have reorganized this section to strengthen the rationale for examining the hypothesis that Gli3-Hand2 cooperation is necessary for mandibular development (subsection “Loss of Gli TFs and Hand2 generates micrognathia and aglossia”). We have also added the triple homozygous conditional KO to Figure 1 to support the subsequent model and added more detail to our anatomical comparisons. Furthermore, we additionally tempered our language surrounding the GO terms associated with processes impacted in these mutants (Figure 2K, L; subsection “Gli2, Gli3, and Hand2 are co-expressed in NCC-derived populations associated with skeletal and glossal progenitors”).